# CoralVQA: A Large-Scale Visual Question Answering Dataset for Coral Reef Image Understanding

**Hongyong Han[1], Wei Wang[1]***, **Gaowei Zhang[1], Mingjie Li[2,3], Yi Wang[1,4,5]**

[1]Beijing University of Posts and Telecommunications
[2]Technology Innovation Center for South China Sea Remote Sensing,
Surveying and Mapping Collaborative Application, Ministry of Natural Resources
[3]South China Sea Development Research Institute, Ministry of Natural Resources
[4]Inspur Computer Technology Co., Ltd
[5]Shandong Key Laboratory of Advanced Computing
{hanhongyong, weiwang, zhanggaowei, yiwang}@bupt.edu.cn, lmj_21@163.com

## Abstract

Coral reefs are vital yet vulnerable ecosystems that require continuous monitoring to support conservation. While coral reef images provide essential information in coral monitoring, interpreting such images remains challenging due to the need for domain expertise. Visual Question Answering (VQA), powered by Large Vision-Language Models (LVLMs), has great potential in user-friendly interaction with coral reef images. However, applying VQA to coral imagery demands a dedicated dataset that addresses two key challenges: domain-specific annotations and multidimensional questions. In this work, we introduce CoralVQA, the first large-scale VQA dataset for coral reef analysis. It contains 12,805 real-world coral images from 67 coral genera collected from 3 oceans, along with 277,653 question-answer pairs that comprehensively assess ecological and health-related conditions. To construct this dataset, we develop a semi-automatic data construction pipeline in collaboration with marine biologists to ensure both scalability and professional-grade data quality. CoralVQA presents novel challenges and provides a comprehensive benchmark for studying vision-language reasoning in the context of coral reef images. By evaluating several state-of-the-art LVLMs, we reveal key limitations and opportunities. These insights form a foundation for future LVLM development, with a particular emphasis on supporting coral conservation efforts.

## 1 Introduction

Coral reefs are among the most biodiverse ecosystems, supporting over a quarter of known marine species (Knowlton et al., 2010; Fisher et al., 2015). They provide ecological value, support coastal economies through fisheries and tourism, and protect shorelines from storms and erosion (Mumby et al., 2008; Bitterwolf et al., 2024). However, human activities and climate change are causing unprecedented global declines (Steffen et al., 2007; Hoegh-Guldberg et al., 2019). Effective global coral reef conservation depends on continuous monitoring of benthic communities, with coral reef images serving as a direct and essential source of information. For instance, experts interpret these images to identify species, evaluate their health, and extrapolate findings to downstream systems. However, the capacity to derive insights from coral images is largely restricted to specialists in marine science. Coral reef monitoring involves dynamic, multiple tasks–including coral recognition, health diagnostics (*e.g.,* bleaching severity), and habitat quality assessment (*e.g.,* algal symbiosis), all of which demand extensive domain expertise for reliable image interpretation (Beijbom et al., 2012).

---

*Corresponding author

39th Conference on Neural Information Processing Systems (NeurIPS 2025) Track on Datasets and Benchmarks.

This barrier restricts the range and variety of problems that can be addressed with coral images (*e.g.,* coral studies in developing countries), and limits the number of potential users (*e.g.,* conservation practitioners, science educators). Therefore, there is a pressing need for new automatic ways that can extract relevant information from coral images without requiring specialized expertise.

The Visual Question Answering (VQA) task–originally developed in computer vision as a user-friendly way for image-based queries–could help bridge this expertise gap (Antol et al., 2015; Wu et al., 2017; Kafle, Kanan, 2017; Lobry et al., 2020; Zheng et al., 2023; Li et al., 2024). For example, given a coral reef image (the corresponding coral reef image is located in the third row, fourth column in Figure 1) and a question ("Is the coral genus in the upper left corner susceptible to bleaching?"). VQA aims to give the correct answer ("Yes"). This transforms specialized ecological assessment into accessible, high-level semantic information. Recent advances in large vision-and-language models (LVLMs) (He et al., 2020; Liu et al., 2023; Bai et al., 2023; Li et al., 2023a; Chen et al., 2024) have demonstrated state-of-the-art performance across various VQA tasks for natural images. However, their extension to the coral reef domain remains limited, primarily due to the absence of comprehensive, high-quality VQA datasets tailored to coral conservation. The construction of coral reef VQA datasets presents substantially greater challenges compared to general-domain VQA, due to two critical factors. (1) **Domain-specific annotations**. While advances in autonomous underwater vehicles have enabled the large-scale collection of coral reef images, existing annotations are inadequate for supporting VQA dataset development. Key issues include: inconsistent label standards–some corals are labeled by morphology, others by genus, and some at the family level; and inclusion of non-coral categories (*e.g.,* sand and sediments). (2) **Multidimensional questions**. Current VQA datasets for natural images primarily focus on generic visual concepts (*e.g.,* animals, vehicles) and basic attributes (*e.g.,* quantity, color). This domain-general characteristic makes question-answer pair generation relatively straightforward for non-experts. However, creating coral VQA datasets requires interdisciplinary expertise in both visual content and textual questions. Additionally, to support coral reef conservation, such a dataset must include open-ended question-answer pairs that span multiple ecological and health-related dimensions, such as coral condition, growth status, and symbiotic relationships.

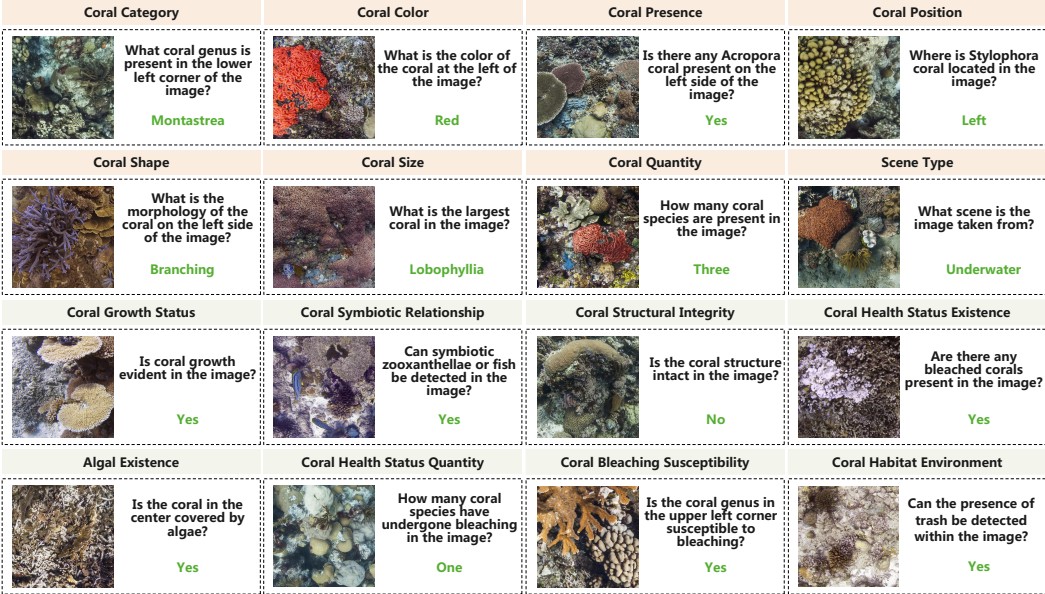

Figure 1: Example of coral reef image and corresponding question-answer pairs across 16 dimensions in CoralVQA dataset. The basic visual attributes of coral reef images are denoted by light orange boxes, while ecological and health-related attributes of coral reef images are indicated by light green boxes.

To address the above challenges, our interdisciplinary team introduces CoralVQA, a large-scale dataset for vision-language understanding of coral reef images, developed through close collaboration with marine biologists possessing extensive expertise in coral conservation. Through data collection,

label cleaning and re-annotating, CoralVQA consists of 12,805 high-quality real coral images from multiple marine regions across 3 oceans, including the Atlantic, Indian Ocean, and Pacific Ocean. All coral instances are re-annotated at the genus level, in accordance with the biological taxonomy hierarchy (*Kingdom-Phylum-Class-Order-Family-Genus*), covering 67 distinct genera from 20 families. CoralVQA encompasses a comprehensive and diverse collection of questions, systematically organized into two key groups: **basic visual interpretation**; **ecological and health-oriented assessment**. Each group comprises eight distinct dimensions. We benchmark several state-of-the-art LVLMs and find that CoralVQA presents novel challenges for coral-specific vision-language reasoning. Notably, we observe a significant drop in model performance when answering questions involving images from previously unseen ocean regions or those requiring complex reasoning, such as assessing bleaching coverage. Some examples from CoralVQA are shown in Figure 1. The key contributions of our work are summarized as follows.

- To the best of our knowledge, CoralVQA is the first large-scale VQA dataset dedicated to coral reef understanding. It contains 12,805 real coral images across 67 genera from 3 oceans, and 277,653 question–answer pairs from 16 dimensions.
- We design a semi-automatic vision-language coral data collection pipeline, which includes six steps: dataset collection, label cleaning and re-annotating, attribute extraction, question generation, and human verification. Our pipeline can be widely applied to other marine domains.
- We systematically evaluate the performance of several state-of-the-art LVLMs on CoralVQA across multiple coral-related tasks, serving as a baseline and highlighting opportunities for future research.

## 2 Related Works

Coral datasets are fundamental for studying coral reef conservation and assessing coral reef health. Existing works on coral datasets mainly focus on classification or segmentation. For example, Tasmania Coral Point Count (TasCPC) (Meyer et al., 2011) contains 1,258 AUV-captured benthic images across 13 underwater object categories (*e.g.*, corals, sand, rock), though coral taxonomic details are unavailable. RSMAS (Shihavuddin et al., 2013) is composed of 766 image patches (256×256 pixels) from 8 coral genera, collected by divers from the University of Miami. Benthoz15 (Bewley et al., 2015), collected by AUVs from Australian, includes 9,874 labeled images. Despite having 148 categories, only two coral genera follow the Linnaean system. EILAT (Shihavuddin, 2017) provides 1,123 Red Sea image patches (64×64 pixels), which represent four morphological coral classes, along with categories for favid coral, dead coral, sand, and urchin. ATCRC (Rashid, Chennu, 2020), collected from Curaçao, contains 147 hyperspectral images representing 6 coral genera.

HSCR16K (Han et al., 2025) contains 16,659 image patches (224×224 pixels) of 10 genera with rich text knowledge. MLC (Beijbom et al., 2012), collected from the island of Moorea, consists of 2,055 coral reef images, including 5 coral genera and 4 non-coral classes. CoralSCOP (Zheng et al., 2024), an important step in coral segmentation, contains 41,297 images from sources like YouTube but focuses on segmentation and lacks coral taxonomic details. In this work, we present CoralVQA, a multi-regions, multi-genera, and multi-dimensions VQA dataset, accompanied by comprehensive benchmarks under diverse VQA scenarios.

Table 1: The comparisons of existing coral datasets.

| Task | Dataset | Images | Genera | QA pairs |
|---|---|---|---|---|
| Cls. | TasCPC | 1,258 | * | × |
| Cls. | RSMAS | 766 | 8 | × |
| Cls. | Benthoz15 | 9,874 | 2 | × |
| Cls. | EILAT | 1,123 | * | × |
| Cls. | ATCRC | 147 | 6 | × |
| Cls. | HSCR16K | 16,659 | 10 | × |
| Seg./Cls. | MLC | 2,055 | 5 | × |
| Seg. | CoralSCOP | 41,297 | * | × |
| VQA | CoralVQA | 12,805 | 67 | 277,653 |

*: Genera are not following the standard Linnaean system.

## 3 Pipeline

To build CoralVQA, we design a semi-automatic vision-language data pipeline ensuring utility and expert-level quality. It includes six stages (see Figure 2): collecting real-world coral reef images; label

cleaning and re-annotation; extracting enriched visual and ecological attributes; designing question generation prompts; auto-generating question-answer pairs and human verification.

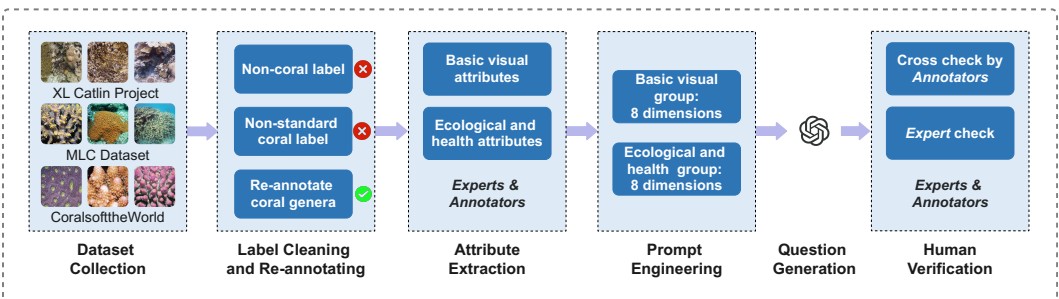

Figure 2: Pipeline for dataset creation. CoralVQA is developed through a six-stage semi-automated vision-language workflow: dataset collection, label cleaning and re-annotating, attribute extraction, prompt engineering, question generation, and human verification.

## 3.1   Dataset Collection

Our CoralVQA collects the real-world coral reef images from 3 sources: MLC dataset (Beijbom et al., 2012), XL Catlin Seaview Survey Project (González-Rivero et al., 2019), and the CoralsoftheWorld database (available at coralsoftheworld.org). The XL Catlin Seaview Survey Project (recording the health of coral reefs worldwide), launched in 2012, includes 11,387 images from the Atlantic, Indian, and Pacific Oceans, providing extensive geographic coverage. To enhance both image quantity and taxonomic coverage, we crawled additional 3,420 images from CoralsoftheWorld, a globally recognized source for scientifically validated coral taxonomy and distribution. This resulted in an initial dataset of 16,862 images. After the quality filter using the Underwater Color Image Quality Evaluation metric (Yang, Sowmya, 2015), the final dataset contains 12,805 high-quality images.

## 3.2   Label Cleaning and Re-annotating

Despite the large volume of collected coral reef images, inconsistent annotation standards and non-coral categories pose challenges for direct downstream use. Therefore, we apply a systematic cleaning and re-annotating process guided by the following criteria: 1) removing the coral labels without following the standard coral biological taxonomic system (*e.g.,* branching Acroporidae and hispidose Acroporidae); 2) eliminating non-coral labels such as sand, fishing gear, and sediments; 3) re-annotating coral instances based on a genus-level coral biological taxonomy hierarchy (from *Kingdom* and *Phylum* to *Family* and *Genus* and *Species*). Marine biologists, led by the fourth author, conducted this step. During the process, 50,200 non-coral annotations and location entries were removed, while 97,395 coral-related annotations and their corresponding locations were re-labeled. In this step, the major issue is inconsistent annotation standards and incorrect coral annotations. To address the problem, we have every image re-examined by three marine scientists after the label cleaning and re-annotating step, and adopt a majority voting approach to determine the final annotations. In fact, only a very small set of images (203 out of 12,805, <2 %) contains errors. The final curated dataset contains 12,805 images spanning 20 families and 67 genera.

## 3.3   Textual Attribute Extraction

We extract enriched attributes from the coral images to facilitate the subsequent automatic generation of question-answer pairs. Guided by the practical needs of coral reef monitoring and conservation, the extracted attributes are organized into two key groups: 1) basic visual attributes and 2) ecological and health-related attributes. Following the paradigm of general VQA tasks, we extract basic visual attributes–such as coral genera, position, and quantity–using automated scripts based on the existing annotation files. In addition, ecological and health-related attributes–including coral health status, growth condition, and symbiotic relationships–are manually annotated for each image.

### 3.4 Prompt Engineering

We carefully develop an automated prompt template to guide GPT-4o in generating detailed question-answer pairs. To enable a comprehensive evaluation and analysis of coral reef images, we formulate questions from two groups spanning 16 distinct dimensions. **Basic visual group** (1) coral category: identify the coral genus based on image position; (2) coral color: identify the coral color at a specific image position; (3) coral presence: determine if coral is present at a given location or if a specific coral exists in the image; (4) coral position: output the position of a specific coral in the image; (5) coral quantity: count the total number of coral genera represented in the image; (6) coral size: identify the dominant or minimal coral genus present in the image; (7) coral shape: describe the coral morphology at a specified position of image; (8) scene type: identify the benthic habitat type in the image (sandy, rocky, or other substrates). **Ecological and health-related group** (1) coral growth condition: determine if the coral exhibits active growth; (2) algal presence: determine if algae cover surrounds the coral (corals and algae exhibit a competitive relationship); (3) coral symbiotic relationship: determine if the coral maintains a stable symbiotic relationship with zooxanthellae; (4) health status existence: determine if coral bleaching or coral disease has occurred at a specific position in the image; (5) coral structural integrity: analyze whether the coral's skeletal structure is intact or if there are signs of damage. (6) health status quantity: count the number of coral genera showing signs of bleaching or disease; (7) bleaching susceptibility: identify if the coral genus belongs to a bleaching-susceptible genus; (8) coral habitat environment: evaluate coral environmental conditions: water clarity, presence of debris, and potential contaminants. To enhance the diversity of the generation question, we carefully design multiple sets of prompts with varying language styles and semantic content for coral images from different data sources.

### 3.5 Question-Answer Generation

We utilize the GPT-4o API to automatically generate question-answer pairs. Existing VQA data generation approaches typically rely on feeding textual inputs into large language models. In contrast, we leverage OpenAI's multimodal capabilities by uploading both textual attributes and coral images through the GPT-4o image API, enabling more authentic and context-aware question–answer generation grounded in visual content. Specifically, based on the image coordinate system, we first divide each image into a $3 \times 3$ grid. Using the coral coordinate annotations, we determine the corresponding grid region for each coral instance. The GPT-4o API then automatically assigns directional indicators (*e.g.,* upper left, center, lower right) to support the generation of location-related questions in downstream tasks. To ensure diversity in question generation, we constrain each question type to appear no more than three times per image. Additionally, we employ the GPT-4o API to perform diversity-oriented paraphrasing, generating questions that differ in linguistic expression while retaining similar semantic intent. We further iteratively refine the prompt templates to better capture domain-specific terminology and improve the overall quality of generated question–answer pairs in the context of coral reef understanding. In this step, the major issue is about the reproducibility issues of GPT-4o generated question-answer pairs. To address the issue, we set the lower temperature parameter of 0.3 and incorporate targeted prompt phrases that explicitly require definitive answers. We randomly sampled 1,000 questions, with each question being answered 10 times by GPT-4o. The average number of consistent answers was 8.2 times. Although GPT-4o generated answers exhibit minor inconsistencies, since all question answers undergo multiple rounds of manual verification, the impact of answer inconsistencies is further mitigated.

### 3.6 Human Verification

Even with carefully designed prompts, errors persist in the GPT-4o API's question–answer pairs. To address these concerns, we perform a three-stage process: 1) manual verification, 2) cross-checking, and 3) expert sampling inspection. In the first stage, twelve students with backgrounds in marine science are recruited to verify the coral image and question-answer pairs manually and fix the errors resulted from using GPT-4o. Each of them needed to deal with roughly 1,000 images. In total, there were 12 subsets of images and corresponding question-answer pairs. Corrections are made based on the following criteria: removing hallucinated questions unrelated to the image; fixing incorrect spatial references within questions; revising uncertain and inaccurate answers. After the first stage, most issues are effectively resolved. To further enhance dataset quality, we implement a cross-checking procedure wherein each verified question-answer pair is reviewed by a second annotator. Each student

cross-checked the subset of images verfied by another student. If the consistency of cross-checking is lower than 95% in any specific subset, a third inspector was assigned to check these inconsistencies again and perform necessary fixes. In the third phase, domain experts inspected 10% images and corresponding question-answer pairs randomly choosen from each subset. If the accuracy of a subset's sample was lower than 95% (judged by the assigned expert), that subset had to be re-verified again. We compiled statistics on the manual pruning of GPT-4o responses. During the human verification stage, we modified 13.4% of the questions generated by GPT-4o and 56.8% of the answers generated by GPT-4o required manual pruning.

## 4 Dataset Statistics

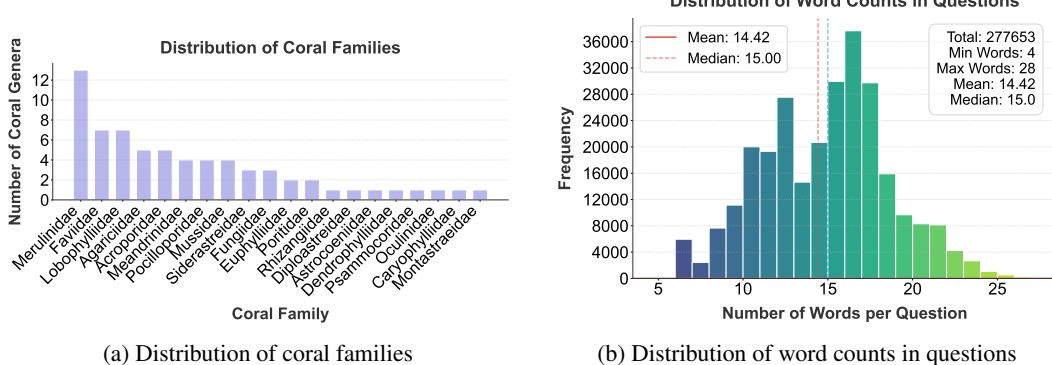

(a) Distribution of coral families        (b) Distribution of word counts in questions

Figure 3: (a) The distribution of coral genera with different coral families in the CoralVQA dataset. (b) Distribution of the word length of questions.

### 4.1 Image Statistics

The CoralVQA dataset contains 12,805 underwater coral reef images and Figure 1 shows some examples. Most of the images are high resolution, with an average pixel width of 1,350 and a pixel height of 1,280. Based on the locations of coral reef survey points, the coral reef images are collected from multiple marine regions across three oceans. These regions include Island of Moorea (MLC), the Atlantic (ATL), the Indian Ocean and Chagos Archipelago (IND_CHA), the Indian Ocean and Maldives (IND_MDV), the Pacific Ocean and USA (PAC_USA), the Pacific Ocean, Indonesia and Philippines (PAC_IDN_PHL), the Pacific Ocean and Solomon Islands (PAC_SLB), the Pacific Ocean and Taiwan (PAC_TWN), and the Pacific Ocean and Timor-Leste (PAC_TLS). The geographic diversity of marine regions significantly enhances the taxonomic richness of coral genera within the dataset, while also enabling cross-regional evaluation of large multimodal models. CoralVQA, comprising 67 coral genera across 20 families, represents the most taxonomically diverse coral dataset to date. The distribution of these genera across their corresponding families is illustrated in Figure 3a. Our code and dataset are publicly available at `https://huggingface.co/datasets/CoralReefData/CoralVQA/tree/main`.

### 4.2 Questions-Answers Statistics

CoralVQA contains 277,653 question-answer pairs with 16 question types. The question types in the dataset are categorized into two main groups: (1) basic visual attributes and (2) ecological and health-related attributes. On average, each coral reef image has 21.6 questions. Figure 3b illustrates the distribution of question lengths in terms of words, with each question containing an average of 14.4 words. Out of the 16 question types, eight are open-ended, including coral category, color, position, quantity, size, shape, scene type, and health status quantity. These open-ended questions make up 42.1% of all question-answer pairs, totaling 116,960 questions. The diversity of answers to open-ended questions and the variety of visual features in coral reefs present significant challenges for visual language models in coral reef VQA tasks. The remaining question types consist of closed-ended "yes/no" questions, with 73,095 questions having "yes" responses and 87,598 with "no" responses. The distribution of different question types is shown in Figure 4.

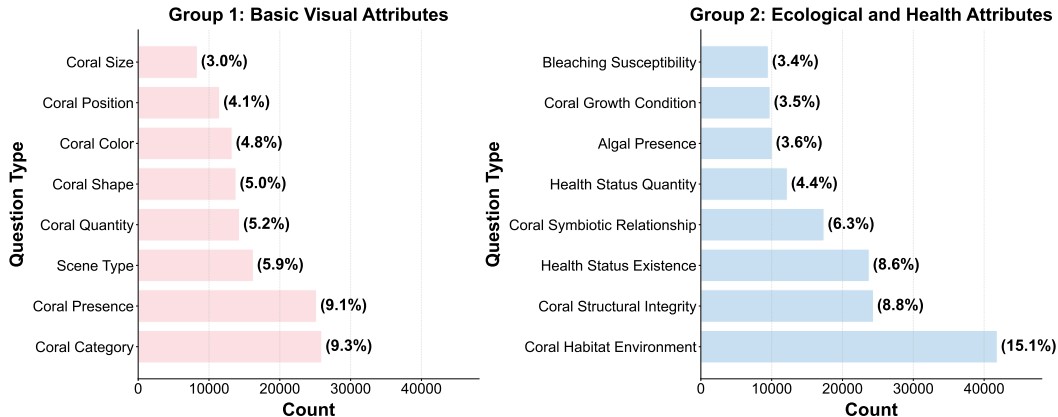

Figure 4: Distribution of two question groups.

# 5 VQA Benchmark Performances

## 5.1 Implementation Details

### 5.1.1 Models

To establish benchmark results, we conduct evaluations on six representative large vision-language models (LVLMs). Among them, the open-source models are Mini-Gemini(7B) (Li et al., 2023b), Qwen2.5VL(7B) (Bai et al., 2023), BLIP3 (Le Xue, 2024), and InternVL2.5(8B) (Chen et al., 2024). In addition, we also include two leading closed-source models, GPT-4o (Hurst et al., 2024) and Claude3.5 Haiku (Anthropic, 2024). For the Mini-Gemini, we utilize the OpenCLIP ConvNext (Il-harco et al., 2021) as the auxiliary vision encoder and use a 2-layer MLP with GeLU activation for projecting visual features. Following the BLIP3, we utilize Microsoft's Phi3 (Abdin et al., 2024) as the base language model and use Google's SigLIP-SO400M (Zhai et al., 2023) as the vision encoder. Other architectures adhere to the original models. For a fair comparison, we initialize all open-source models with their own pre-trained weights. Additional training details are provided in the appendix.

### 5.1.2 Coral VQA Tasks

We divide CoralVQA into three non-overlapping subsets to enable comprehensive evaluation: a training dataset, a testing dataset, and a cross-region dataset (marine region near Hawaii). Detailed statistics are presented in Table 2. We then establish three tasks related to coral reef conservation. (1) **VQA on test dataset**: Evaluation on the test dataset to establish standard performance benchmark. (2) **VQA on cross-region dataset**: Evaluation on the cross-region dataset, designed to examine model generalization to unseen regions. (3) **VQA on bleaching-coverage dataset**: Evaluation on our novel small-scale bleaching-coverage dataset (detailed in Section 5.4), which tests the model's ability to perform a complex reasoning task. We fine-tune all open-source models on the training set and subsequently evaluate their performance on different tasks.

Table 2: Statistics of train, test, and cross-region dataset. "Q" and "A" denote question and answer.

| Items | Train | Test | Cross |
|---|---|---|---|
| Images | 10,537 | 1,274 | 994 |
| QA pairs | 226,726 | 27,984 | 22,943 |
| Avg words per Q | 14.45 | 13.25 | 15.58 |
| Avg words per A | 1.12 | 1.01 | 1.04 |

## 5.2 VQA Results on Test Dataset

**Settings and Evaluation** Based on the question types in the CoralVQA dataset, we divide the testing set into 16 subsets and measure the model's performance using the accuracy as the metric. To gain a more complete understanding of the model's performance, we calculate the average accuracy separately from two groups: basic visual attributes, and ecological and health attributes. We utilize the GPT-4o API to evaluate whether the semantics of the ground-truth answer from CoralVQA match

those of answers from the models. For a comprehensive description of the calculation methods, please refer to the appendix.

**Results** As shown in Table 3, the following three conclusions can be drawn. First, compared to other methods, InternVL2.5 achieves superior average performance across both groups. In addition, we further perform the experiments with two closed-source models: GPT-4o and Claude-3.5 Haiku. Although existing closed-source models have achieved good performance on general tasks, they perform poorly on coral-related tasks. Second, the accuracy of all models on open-ended questions is significantly lower than on closed-ended questions, with a performance gap exceeding 10%. For instance, the accuracy of InternVL2.5 for scene-type questions is 14.74% higher than for category-related questions. Similarly, Qwen2.5VL shows a 39.70% higher accuracy in addressing scene-type questions compared to category-related questions. These results suggest that open-ended questions often require domain-specific knowledge of coral categories and morphological characteristics, while current vision-language models pre-trained on general image-text datasets lack specialized knowledge in this area. Third, all methods show lower accuracy on questions regarding coral shape and quantity. In coral reef images, multiple coral genera are often distributed in interspersed patterns, with some genera being partially occluded. The limited ability of current visual-language models to extract textural, edge, and morphological features from coral reef images leads to enumeration errors, including both missed and duplicate counts.

Table 3: Visual question answering performance on test dataset. Boldface indicates the best performance.

| Methods | Category | Presence | Quantity | Color | Position | Size | Shape | Scene | All |
|---|---|---|---|---|---|---|---|---|---|
| GPT-4o | 71.35 | 50.38 | 7.04 | 70.09 | 43.13 | 54.26 | 57.47 | 85.83 | 54.94 |
| Claude3.5 Haiku | 69.27 | 52.26 | 5.92 | 43.08 | 48.37 | 62.63 | 38.36 | 47.54 | 45.93 |
| Mini-Gemini(FT) | 8.72 | 28.86 | 6.98 | 2.01 | 7.23 | 4.47 | 3.18 | 40.65 | 12.76 |
| BLIP3(FT) | 69.43 | 62.66 | 14.90 | **87.72** | 36.76 | 1.30 | 34.72 | 95.47 | 50.37 |
| Qwen2.5VL(FT) | 55.00 | 70.43 | 14.90 | 70.68 | 46.18 | 59.45 | 57.47 | 94.70 | 58.60 |
| InternVL2.5(FT) | **81.69** | **79.45** | **35.54** | 78.27 | **62.18** | **74.46** | **62.85** | **96.43** | **71.35** |
| **Methods** | **Growth** | **Algal** | **Presence** | **Quantity** | **Integrity** | **Susceptibility** | **Environment** | **Symbiosis** | **All** |
| GPT-4o | 52.81 | 42.88 | 25.21 | 14.78 | 33.68 | 28.68 | 61.84 | 87.48 | 43.42 |
| Claude3.5 Haiku | 44.00 | 38.48 | 42.96 | 34.11 | 18.05 | 30.64 | 62.00 | 30.64 | 37.61 |
| Mini-Gemini(FT) | 38.74 | 27.88 | 30.61 | 20.96 | 37.26 | 38.71 | 24.82 | 53.34 | 34.04 |
| BLIP3(FT) | 88.09 | 61.67 | **94.51** | 63.47 | **96.54** | **80.92** | 74.83 | 82.60 | 80.32 |
| Qwen2.5VL(FT) | 78.52 | 58.18 | 90.56 | 60.50 | 72.56 | 74.37 | 72.00 | 72.45 | 72.39 |
| InternVL2.5(FT) | 86.96 | **65.45** | 88.73 | **65.25** | 95.90 | 80.04 | **80.20** | **88.81** | **81.42** |

Table 4: Visual question answering performance on cross-region dataset. Boldface indicates the best performance. The "-" denotes that missing question-answer data for this dimension.

| Methods | Category | Presence | Quantity | Color | Position | Size | Shape | Scene | All |
|---|---|---|---|---|---|---|---|---|---|
| Mini-Gemini(FT) | 4.93 | 16.88 | **9.59** | 1.83 | 5.58 | 2.37 | 2.33 | 23.36 | 8.36 |
| BLIP3(FT) | **25.21** | **29.26** | 1.37 | **12.82** | 9.88 | 17.00 | 8.35 | 30.71 | 16.83 |
| Qwen2.5VL(FT) | 20.33 | 26.17 | 1.11 | 8.41 | 12.80 | **25.49** | 12.48 | 28.84 | 16.95 |
| InternVL2.5(FT) | 24.09 | 26.45 | 2.14 | 10.07 | **14.43** | 23.12 | **14.45** | **35.25** | **18.75** |
| **Methods** | **Growth** | **Algal** | **Presence** | **Quantity** | **Integrity** | **Susceptibility** | **Environment** | **Symbiosis** | **All** |
| Mini-Gemini(FT) | 18.65 | 15.48 | 18.84 | **17.13** | 17.38 | - | 11.78 | 20.64 | 17.13 |
| BLIP3(FT) | **32.35** | 26.03 | **32.06** | 8.80 | **31.07** | - | **38.03** | 33.80 | **28.88** |
| Qwen2.5VL(FT) | 31.82 | 24.10 | 30.12 | 2.78 | 26.04 | - | 33.50 | 31.54 | 25.70 |
| InternVL2.5(FT) | 31.93 | **26.21** | 31.20 | 6.25 | 29.89 | - | 36.58 | **35.71** | 28.25 |

### 5.3 VQA Results on Cross-Region Dataset

**Settings and Evaluation** Coral genera exhibit considerable intra-class variation in their composition, color, and morphology from different marine regions. Following the previous evaluation metrics, we adopt the accuracy as the metric to assess the model's generalization performance on the cross-region dataset. Similar to the previous setting, we employ the GPT-4o API to assess the semantic consistency between cross-region dataset ground-truth answers and model-generated answers.

**Results** As shown in Table 4, compared to the results on the standard test dataset, the performance of nearly all models decreased by more than 30% in both groups. For instance, InternVL2.5 achieves only 2.14% accuracy on coral quantity-related questions and shows more than 10% drops in accuracy

for questions related to coral position, shape, and color. These results indicate that existing visual-language models exhibit limitations in generalization when handling coral images from diverse ecological or geographical environments.

## 5.4 VQA Results on Bleaching-Coverage Dataset

**Settings and Evaluation** Evaluating the proportion of coral bleaching is essential for determining the extent of impacted ecosystems and the number of affected coral populations. Existing methods of coral bleaching coverage assessment typically rely on image segmentation to obtain the bleached coral area and calculate the coral bleaching coverage ratio. However, these methods heavily rely on domain knowledge and involve multiple steps, making them limited in application and time-consuming. In this context, VQA offers a promising alternative. The evaluation of coral bleaching rates through VQA presents significant challenges. This process requires not only pixel-level semantic segmentation and quantitative analysis of coral regions in images, but also a comprehensive understanding of both the spatial distribution and the visual details within coral images. In this section, we introduce a new VQA dataset called Bleaching-Coverage. Specifically, we collect 309 bleached coral reef images from the CoralVQA dataset and build the 309 question-answer pairs about coral bleaching coverage. We employ the LabelMe tool to segment the regions of bleached coral genera and calculate the proportion of the bleached area to the entire surveyed area, thereby obtaining the ground-truth. We utilize the MAE (Mean Absolute Error) and MASE (Mean Absolute Scaled Error) to evaluate the error between the output from VQA models and the ground truth.

**Results** The mean absolute error (MAE) of Qwen2.5VL is 0.1124, whereas that of InternVL2.5 is 0.0818. The MASE for Qwen2.5VL and InternVL2.5 are 1.2326 and 0.8967, respectively. Compared with Qwen2.5VL, InternVL2.5 has lower values of MAE and MASE, indicating better generalize ability for complex reasoning tasks. However, all methods still exhibit substantial prediction errors. Furthermore, Mini-Gemini and BLIP3 fail to effectively comprehend the task-specific questions, often generating unrelated answers. These results suggest that existing visual-language models still face significant challenges in these complex tasks. In addition, we visualize the bleaching regions of corals predicted by InternVL2.5 in the appendix, which further illustrates the limitations of existing vision-language models in assessing bleaching coverage.

## 5.5 Further Discussion

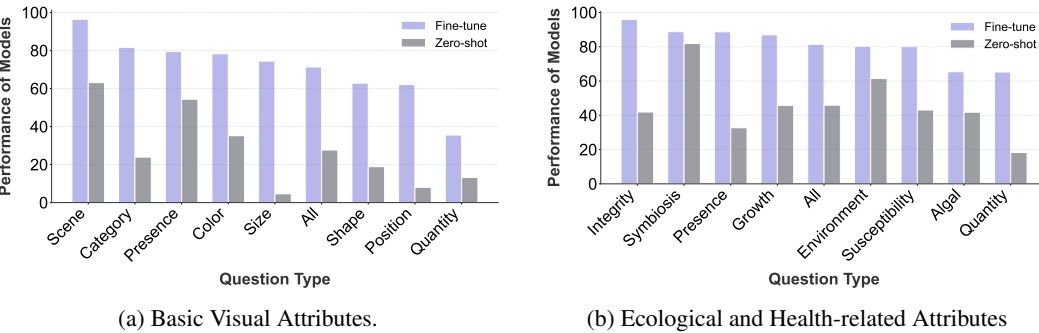

(a) Basic Visual Attributes.

(b) Ecological and Health-related Attributes

Figure 5: Performance comparison between fine-tuned and zero-shot evaluation on InternVL2.5 model. "All" represents the average accuracy across all question types.

**Zero-shot Evaluation** To further evaluate the real performance of existing visual-language models pre-trained on general image-text datasets in coral reef image understanding, we perform zero-shot evaluation on the InternVL2.5 models. Following the same evaluation method as before, we utilized the GPT-4o API to evaluate answer accuracy. As shown in Figure 5, two conclusions can be drawn. First, compared to the fine-tuning strategy, the average performance of zero-shot evaluation for InternVL2.5 shows varying degrees of degradation. For instance, InternVL2.5's zero-shot evaluation shows performance decreases of 43.68 % and 35.54 % in the basic visual attributes, and ecological and health-related attributes, respectively. The result also indicated the effectiveness of fine-tuning strategies from the vision-language model. Second, for questions related to coral size, position, and quantity, the InternVL2.5's question-answering accuracy falls below 20 %. The result indirectly

reflects the limited capability of current vision-language models in extracting texture, edge, and morphological features from coral reef images. More detailed results are provided in the appendix. **Case Study** To further investigate the reason behind the model's lower performance on questions related to coral size and shape, we conduct a case study. Focusing on InternVL2.5, as shown in Figure 6, we perform a visual analysis of its decision regions when answering questions about coral size and quantity. We design the following prompts to guide the model's responses: "What is the genus of the largest hard coral in the image? Provide a brief response and output the location coordinates." "How many coral genera are present in the image? Give a concise answer and specify their respective location coordinates." For coral size-related questions, the InternVL2.5 model incorrectly identifies non-target categories as the largest coral genus–mistakenly selecting regions labeled as other organisms, while the correct answer is the genus *Porites*, located at the top of the image. This mistake is mainly due to the model's lack of domain-specific knowledge, which hinders its ability to distinguish between coral genera and similar-looking categories such as algae. For coral quantity-related questions, the InternVL2.5 model is unable to identify the positions of individual coral genera, leading to incorrect counts. These results highlight the model's limitations in spatial reasoning, indicating that it lacks the fine-grained ecological understanding necessary for accurate differentiation among visually similar coral types. Furthermore, they demonstrate the necessity of developing novel frameworks for integrating marine biological knowledge into visual-language models to enhance their performance on coral analysis tasks.

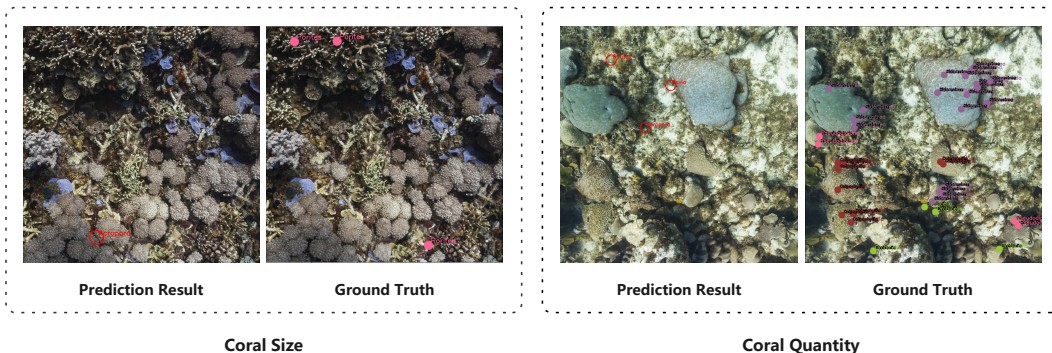

Figure 6: Visualization results. The hollow circles on the left indicate predicted positions of the model, while the solid circles on the right represent positions of the ground truth.

## 6    Conclusion and Future Works

In this paper, we introduce Visual Question Answering as an accessible way of extracting high-level semantic information from coral image data. To this end, we build a coral VQA dataset that includes 12,805 real coral reef images of 67 coral genera and 277,653 question-answer pairs of 16 dimensions. We further introduce a semi-automatic data collection pipeline, empowering researchers to extend and customize the coral dataset, and even generalize the framework to other marine ecosystems. Our benchmarking of several state-of-the-art LVLMs reveals significant challenges in coral VQA tasks, particularly when handling unseen marine regions and complex logical reasoning questions. These findings provide critical insights and establish new research directions for advancing vision-language models in coral research and protection.

## 7    Acknowledgments

This work is partially supported by National Natural Science Foundation of China under grants 62076232 and 62172049, BUPT Excellent Ph.D. Students Foundation under grant CX20251006, Science and Technology Development Foundation of South China Sea Bureau, Ministry of Natural Resources under grant 230208, Guangdong S&T Programme under grant 2025B1111130002, and Key Program of Marine Economy Development Special Foundation of Department of Natural Resources of Guangdong Province under grant GDNRC[2020]012. We sincerely thank the marine biologists in the fourth author's research team (Nansha Islands Coral Reef Ecosystem National Observation and Research Station) for their help on data collection and processing.

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
