# CoralVQA: A Large-Scale Visual Question Answering Dataset for Coral Reef Image Understanding

**Hongyong Han[1], Wei Wang[1]\*, Gaowei Zhang[1], Mingjie Li[2,3], Yi Wang[1,4,5]**
[1]Beijing University of Posts and Telecommunications
[2]Technology Innovation Center for South China Sea Remote Sensing,
Surveying and Mapping Collaborative Application, Ministry of Natural Resources
[3]South China Sea Development Research Institute, Ministry of Natural Resources
[4]Inspur Computer Technology Co., Ltd
[5]Shandong Key Laboratory of Advanced Computing
{hanhongyong, weiwang, zhanggaowei, yiwang}@bupt.edu.cn, lmj_21@163.com

## 1 CoraVQA Data Documentation and Intended Uses

### 1.1 Overview

CoralVQA contains 12,805 real-world coral images from 67 coral genera collected from 3 oceans, along with 277,653 question-answer pairs that comprehensively assess ecological and health-related conditions. CoralVQA provides a comprehensive benchmark for studying vision-language reasoning in the context of coral reef images. This section offers detailed documentation of the CoralVQA dataset to guarantee the transparency, replicability, and ethical application.

### 1.2 Organization of Data and Code

The CoralVQA dataset and code are organized as follows.

```
root/
|__ CoralVQA_Image.zip
|__ CoralVQA_train.jsonl
|__ CoralVQA_test.jsonl
|__ CoralVQA_cross-region.jsonl
|__ Code
|__ __ BLIP3.zip
|__ __ InternVL.zip
|__ __ Mini-Gemini.zip
|__ __ Qwen2.5-VL.zip
```

Detailed file documentation is described in the following.

- **CoralVQA_Image.zip:** The file contains 12,805 real-world coral images from 67 coral genera.
- **CoralVQA_train.jsonl:** The JSONL file consists of all question-answer pairs of the training datasets. The JSONL file is organized following the LLAVA data format. Each row is a question-answer pair. Each question-answer pair contains an ID, image path, conversations, and response to conversations.
- **CoralVQA_test.jsonl:** The JSONL file comprises 27,984 question-answer pairs. Each row contains four components: ID, image path, question, and question type.

---

\*Corresponding author

- **CoralVQA_cross-region.jsonl:** The cross-region question-answer dataset is designed to evaluate the generalization capability of vision-language models. The file contains 22,943 question-answer pairs in standard JSONL format.

- **Code:** The folder contains both fine-tuning and evaluation code for all methods, including Mini-Gemini(7B) Li et al. (2023), Qwen2.5VL(7B) Bai et al. (2023), BLIP3 Le Xue (2024), InternVL2.5(8B) Chen et al. (2024).

## 1.3 Intended Uses

The intended use of the CoralVQA dataset is to support the development, evaluation, and benchmarking of visual-language models and systems that integrate visual perception and natural language understanding in the context of coral reef images. CoralVQA is mainly designed for academic and research settings, specifically for:

- The CoralVQA is designed to facilitate research in fine-grained coral recognition, domain-specific reasoning, and interactive ecological monitoring. The CoralVQA dataset facilitates the development of models capable of automated natural language question-answering regarding underwater environments, coral genera identification, and coral reef health assessment.

- The CoralVQA serves as a valuable data resource for advancing explainable AI, educational tools in marine science.

## 1.4 Limitations

- **Limitation in Taxonomic Resolution:** The taxonomic resolution of the CoralVQA dataset is at the genus level for corals. However, annotating corals at the species level presents significant challenges. First, different species within the same coral genus exhibit highly similar morphological features, making it difficult even for experts to distinguish them rapidly based on image data alone. Second, species-level annotation requires substantially more time and effort.

- **Annotation Bias:** Despite efforts to ensure high-quality question-answer pairs through cross-validation and expert checks, potential biases may still exist arising from subjective human factors.

## 1.5 Ethical Considerations

- **Data Source Transparency and Legitimacy:** Our CoralVQA collects the real-world coral reef images from 3 sources: MLC dataset Beijbom et al. (2012), XL Catlin Seaview Survey Project González-Rivero et al. (2019), and the CoralsoftheWorld database (available at coralsoftheworld.org). All coral reef images are publicly available at `https://huggingface.co/datasets/CoralReefData/CoralVQA/tree/main`.

- **Sample Bias:** The CoralVQA contains 12,805 real-world coral images collected from 3 oceans and 9 regions, which avoids sample bias from a single region.

- **Use Restrictions:** We encourage responsible and ethical usage of the CoralVQA dataset, particularly for research related to coral reef monitoring and conservation.

## 1.6 Broader Impact

To the best of our knowledge, CoralVQA is the first large-scale VQA dataset dedicated to coral reef understanding. The CoralVQA dataset provides a comprehensive benchmark for developing and evaluating vision-language models oriented towards coral reef monitoring and conservation. On the one hand, vision-language models trained on the CoralVQA dataset can assist researchers in rapidly identifying coral genera, coral growth conditions, and coral health status, thereby enhancing ecological monitoring efficiency. On the other hand, an interactive coral reef visual question answering system can further facilitate public learning of coral reef knowledge.

## 1.7 Declaration of LLM usage

We utilize the GPT-4o API to generate the initial question-answer pairs. Additional details can be found in the main manuscript. All question-answer pairs undergo human cross-validation and expert sampling inspection.

## 1.8 Use Cases

- **Academic Research:** CoralVQA provides a multimodal benchmark that opens new avenues for automated analysis of coral reef images, thereby supporting downstream tasks relevant to coral reef ecological monitoring and conservation. Automated analysis and responses to questions regarding coral genera classification and distribution will further advance marine ecological research and conservation efforts.

- **Model Development and Evaluation:** The CoralVQA dataset, serving as the training dataset and testing dataset, enables the development of vision-language models for coral reef image comprehension.

- **Educational Purposes:** Based on the CoralVQA dataset, interactive coral reef visual question answering systems can be developed, which will promote public education and understanding of coral reef ecosystems.

## 1.9 Distribution Regions of Coral Reef Images

Our CoralVQA collects the real-world coral reef images from 3 sources: MLC dataset Beijbom et al. (2012), XL Catlin Seaview Survey Project González-Rivero et al. (2019), and the CoralsoftheWorld database (available at coralsoftheworld.org). The MLC dataset contains 1,998 coral reef images and is collected from the Island of Moorea. The coral reef images from the XL Catlin Seaview Survey Project are gathered from 8 regions: the Atlantic (ATL), the Indian Ocean and Chagos Archipelago (IND_CHA), the Indian Ocean and Maldives (IND_MDV), the Pacific Ocean and USA (PAC_USA), the Pacific Ocean, Indonesia and Philippines (PAC_IDN_PHL), the Pacific Ocean and Solomon Islands (PAC_SLB), the Pacific Ocean and Taiwan (PAC_TWN), and the Pacific Ocean and Timor-Leste (PAC_TLS). The distribution of coral reef images across eight regions is illustrated in Figure 1a. The region-related information from the CoralsoftheWorld database is unavailable. The spatial heterogeneity across marine regions not only enriches the dataset's coral genus diversity but also facilitates performance evaluation of large multimodal models across different regions.

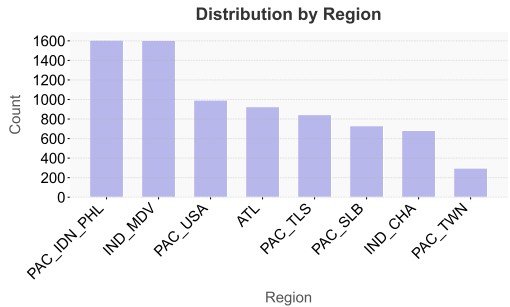

(a) Distribution of coral reef images across different regions.

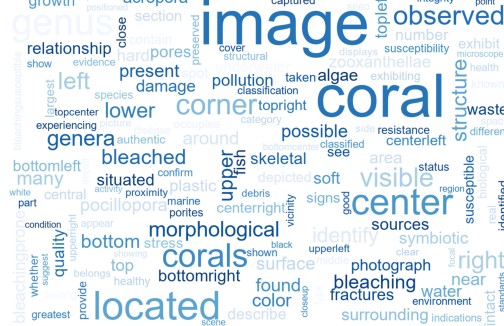

(b) Word cloud of all questions

Figure 1: (a) Distribution of coral reef images across eight regions from the XL Catlin Seaview Survey Project. (b) Word cloud of all questions in the CoralVQA dataset.

## 1.10 Overview of Coral Genera and Corresponding Families

To support broader taxonomic research and biodiversity assessment, our proposed CoralVQA dataset contains 67 coral genera and 20 coral families. Current coral taxonomic systems lack complete standardization, resulting in inconsistent classification of some coral genera and families among various research works. Through collaboration with marine biology experts, we classified existing

Acropora

Goniastrea

Goniopora

Heliofungia

Hydnophora

Isopora

Leptoseris

Lobophyllia

Madracis

Montipora

Orbicella

Pavona

Pocillopora

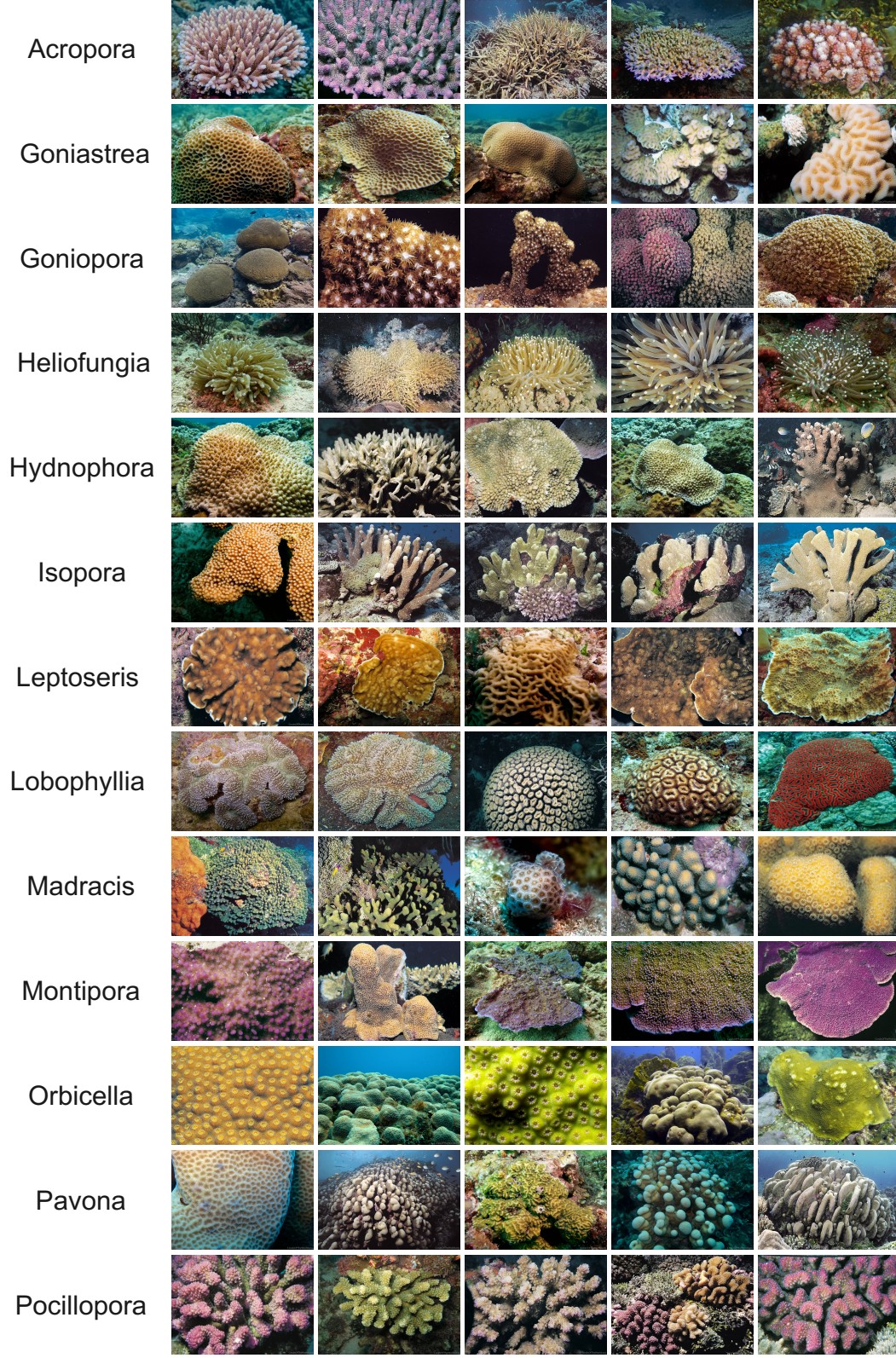

Figure 2: Example of coral reef images.

Table 1: Overview of coral genera and corresponding family

| Family | Genera | Family | Genera |
|---|---|---|---|
| Merulinidae | Cyphastrea; Echinopora; Favites; Goniastrea; Hydnophora; Leptoria; Merulina; Oulastrea; Orbicella; Oulophyllia; Paramontastraea; Platygyra; Montastrea | Poritidae | Goniopora; Porites |
| Lobophylliidae | Acanthastrea; Echinophyllia; Lobophyllia; Lithophyllon; Oxypora; Mycedium; Gyrosmilia | Euphylliidae | Plerogyra; Galaxea |
| Faviidae | Leptastrea; Plesiastrea; Cladocora; Colpophyllia; Favia; Diploria; Manicina | Rhizangiidae | Astrangia |
| Agariciidae | Agaricia; Gardineroseris; Leptoseris; Pachyseris; Pavona | Diploastreidae | Diploastrea |
| Acroporidae | Acropora; Alveopora; Astreopora; Isopora; Montipora | Astrocoeniidae | Stylocoeniella |
| Meandrinidae | Dichocoenia; Eusmilia; Meandrina; Dendrogyra | Dendrophylliidae | Turbinaria |
| Pocilloporidae | Madracis; Pocillopora; Seriatopora; Stylophora | Psammocoridae | Psammocora |
| Mussidae | Isophyllia; Mussa; Mussismilia; Mycetophyllia | Oculinidae | Oculina |
| Siderastreidae | Anomastraea; Siderastrea; Solenastrea | Caryophylliidae | Heterocyathus |
| Fungiidae | Cycloseris; Heliofungia; Herpolitha | Montastraeidae | Montastraea |

coral genera by integrating two taxonomic systems: WoRMS (available at marinespecies.org) and coralsoftheworld (available at coralsoftheworld.org). The overview of coral genera and the corresponding family is illustrated in Table 1. As shown in Figure 2, we present some representative coral reef images of some coral genera.

## 1.11 Prompt

We carefully develop an automated prompt template to guide GPT-4o in generating detailed question-answer pairs. Detailed prompt templates are presented below.

You are a marine ecologist expert at analyzing the coral reef image. You will receive an image along with its corresponding list of visual attributes, formatted as follows:

```
{
  image_id: ...,
  objects:
  [
    { obj_cls: ...,
      obj_coord: ...,
      coral health status:...,
      coral growth condition:...,
      ...
    },
    ...
  ]
}.
```

Your task is to generate a set of 25 question-answer pairs. The questions should cover 16 question types across the following two groups. Basic visual attributes: questions that refer to the coral

category, color, presence, position, quantity, size, shape, and scene type. Ecological and health-related attributes: questions related to coral growth condition, algal presence, coral symbiotic relationship, health status existence, coral structural integrity, health status quantity, coral bleaching susceptibility, and coral habitat environment. For questions whose answers may not be provided in the list, you need to engage in deep reasoning and analysis of the input images to infer the most accurate answer. You are required to output a JSON-formatted file in the following structure:

```
[
 {
        "image_id": "Image Name",
        "question": "Generated Questions",
        "ground_truth": "Your Answer",
        "dataset": "CoralVQA",
        "question_id": "Question ID",
        "type": "Question Type"
    },
  ...
]
```

Here are further important instructions for visual question answering.

1. For each coral reef image, please formulate questions using diverse linguistic styles and content.

2. Questions should yield unambiguous responses, with answers limited to concise expressions of no more than three words.

3. Avoid using expressions such as "unknown", "uncertain", or "indeterminate" when generating question answers. If the answer is uncertain or unable to be determined, remove this question-answer pair.

4. When answering questions about coral quantity, please consider the number of all coral genera present in the image.

5. Please attempt to cover all sixteen question types.

## 1.12 Most-Frequent Words in the Questions

Figure 1b presents the word cloud for all questions in the CoralVQA dataset. Figure 3 and Figure 4 present word clouds of the top 20 most frequent words for each question type in the basic visual attributes, and ecological and health-related attributes, respectively.

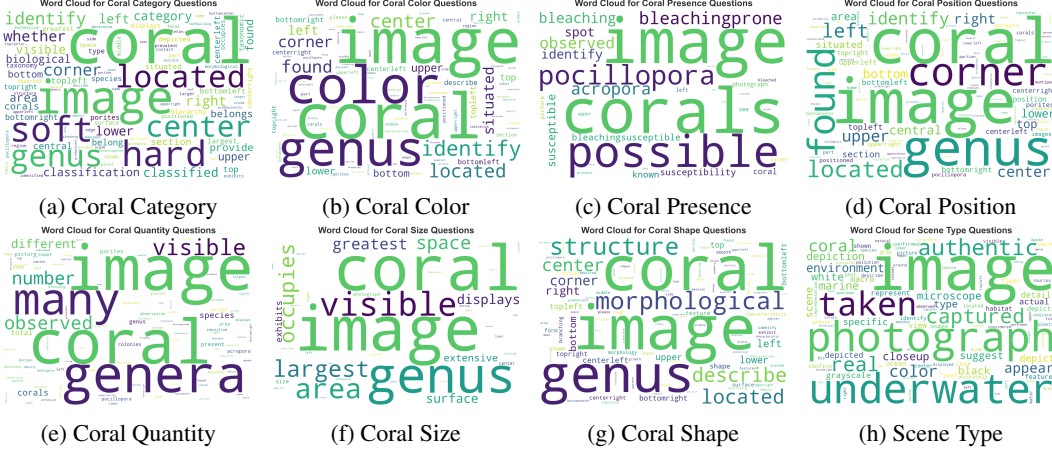

(a) Coral Category  (b) Coral Color  (c) Coral Presence  (d) Coral Position

(e) Coral Quantity  (f) Coral Size  (g) Coral Shape  (h) Scene Type

Figure 3: Word clouds of the top 20 most frequent words for each question type in the basic visual attributes.

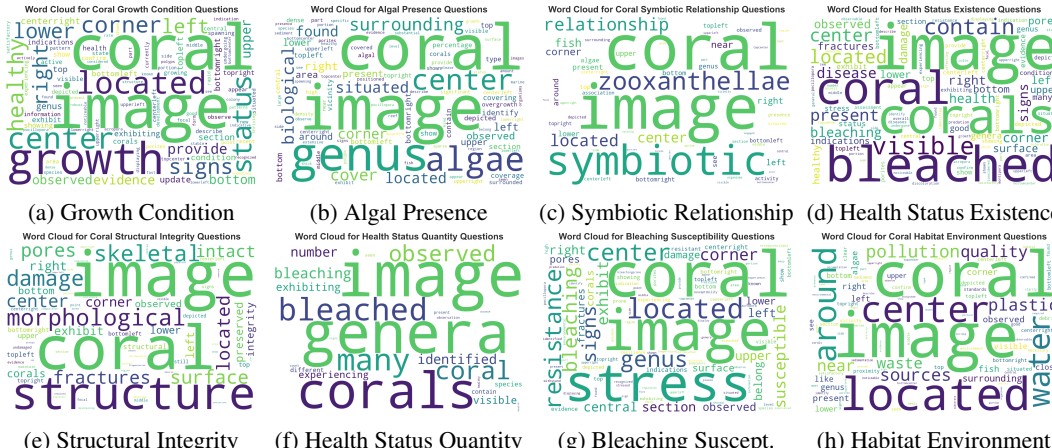

(a) Growth Condition    (b) Algal Presence    (c) Symbiotic Relationship    (d) Health Status Existence

(e) Structural Integrity    (f) Health Status Quantity    (g) Bleaching Suscept.    (h) Habitat Environment

Figure 4: Word clouds of the top 20 most frequent words for each question type in ecological and health-related attributes. "Bleaching Suscept." denotes bleaching susceptibility.

## 2 Experiments

### 2.1 Experiment Setting

All comparison models are trained on single node with 4 NVIDIA H20 GPUs. All models are run in an environment with PyTorch 2.1.0, Python 3.10, and CUDA 12.1. For a fair comparison, all visual-language models are fine-tuned for three epochs with an initial learning rate of 2e-4 and a cosine learning rate decay schedule for optimization.

### 2.2 Evaluate Accuracy with GPT Review

To ensure a robust evaluation, we utilize the GPT-4o API to evaluate whether the semantics of the ground-truth answer from CoralVQA match those of answers from the models. Our carefully designed prompts are as follows: *Question: question, Ground Truth Answer: ground_truth, Predicted Answer: predicted, Does the predicted answer match the ground truth? Answer 1 for match and 0 for not match. Use semantic meaning not exact match. Synonyms are also treated as a match, e.g., zero and 0, 1 and one. Do not explain the reason.*

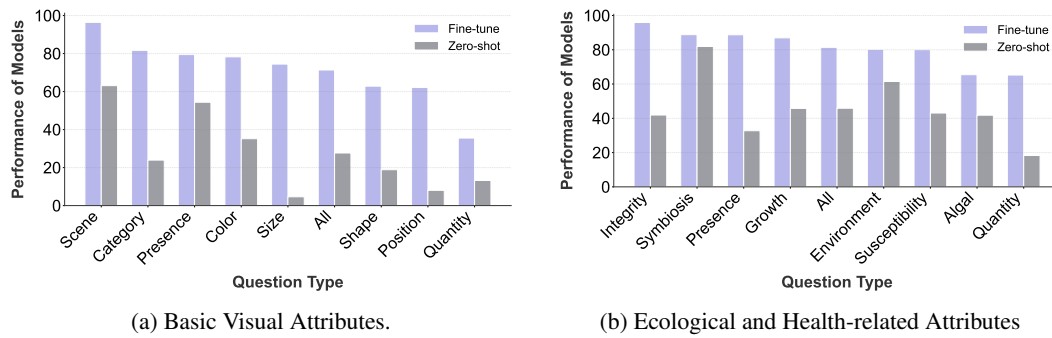

(a) Basic Visual Attributes.      (b) Ecological and Health-related Attributes

Figure 5: Performance comparison between fine-tuned and zero-shot evaluation on InternVL2.5 model. "All" represents the average accuracy across all question types.

### 2.3 Zero-shot Evaluation

To further evaluate the real performance of existing visual-language models in coral reef image understanding, we perform zero-shot evaluation on the Qwen2.5VL and InternVL2.5 models. Following the same evaluation method as before, we utilized the GPT-4o API to evaluate answer accuracy. Two conclusions can be drawn. First, compared to the fine-tuning strategy, the average performance of

zero-shot evaluation for both Qwen2.5VL and InternVL2.5 shows varying degrees of degradation. For instance, InternVL2.5's zero-shot evaluation shows performance decreases of 43.68 % and 35.54 % in the basic visual attributes, and ecological and health-related attributes, respectively. The result also indicated the effectiveness of fine-tuning strategies from the vision-language model. Second, as shown in Figure 5, for questions related to coral size, position, and quantity, the InternVL2.5's question-answering accuracy falls below 20 %. The result indirectly reflects the limited capability of current vision-language models in extracting texture, edge, and morphological features from coral reef images.

Table 2: Visual question answering performance of fine-tuned models on the test dataset.

| Methods | Category | Presence | Quantity | Color | Position | Size | Shape | Scene | All |
|---|---|---|---|---|---|---|---|---|---|
| Qwen2.5VL | 55.00 | 70.43 | 14.90 | 70.68 | 46.18 | 59.45 | 57.47 | 94.70 | 58.60 |
| InternVL2.5 | 81.69 | 79.45 | 35.54 | 78.27 | 62.18 | 74.46 | 62.85 | 96.43 | 71.35 |
| Methods | Growth | Algal | Presence | Quantity | Integrity | Susceptibility | Environment | Symbiosis | All |
| Qwen2.5VL | 78.52 | 58.18 | 90.56 | 60.50 | 72.56 | 74.37 | 72.00 | 72.45 | 72.39 |
| InternVL2.5 | 86.96 | 65.45 | 88.73 | 65.25 | 95.90 | 80.04 | 80.20 | 88.81 | 81.42 |

Table 3: Visual question answering performance of zero-shot evaluation on the test dataset.

| Methods | Category | Presence | Quantity | Color | Position | Size | Shape | Scene | All |
|---|---|---|---|---|---|---|---|---|---|
| Qwen2.5VL | 63.72 | 51.55 | 12.59 | 53.05 | 47.45 | 58.59 | 9.02 | 56.80 | 44.09 |
| InternVL2.5 | 23.94 | 54.39 | 13.22 | 35.19 | 8.03 | 4.62 | 18.88 | 63.11 | 27.67 |
| Methods | Growth | Algal | Presence | Quantity | Integrity | Susceptibility | Environment | Symbiosis | All |
| Qwen2.5VL | 44.93 | 37.58 | 53.73 | 36.43 | 52.89 | 52.89 | 68.63 | 87.62 | 54.34 |
| InternVL2.5 | 45.78 | 41.82 | 32.75 | 18.29 | 41.92 | 43.08 | 61.49 | 81.93 | 45.88 |

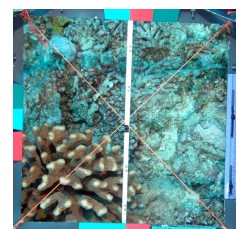

**How many coral genera can be observed in the image?**
BLIP3:Zero  Qwen2.5VL:One  InternVL2.5:One  GT:One

**What is the color of the coral genus located in the bottom-left of the image?**
BLIP3:Brown  Qwen2.5VL:None  InternVL2.5:Brown  GT:Brown

**What is the morphological structure of the coral genus in the bottom-left of the image?**
BLIP3:Massive  Qwen2.5VL:Branching  InternVL2.5:Branching  GT:Branching

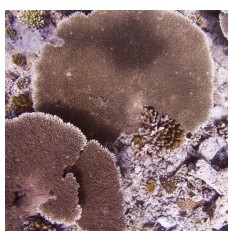

**How many coral genera can be observed in the image?**
BLIP3:Zero  Qwen2.5VL:Two  InternVL2.5:Three  GT:Two

**How many genera of bleached corals can be observed in the image?**
BLIP3:Zero  Qwen2.5VL:Zero  InternVL2.5:One  GT:One

**Which coral genus has the largest visible area in the image?**
BLIP3:None  Qwen2.5VL:Acropora  InternVL2.5:Acropora  GT:Acropora

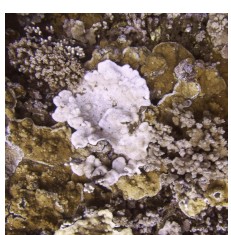

**Where is the bleached Montipora coral located in the image?**
BLIP3:Yes  Qwen2.5VL:Yes  InternVL2.5:Yes  GT:Center

**Which coral genus can be found in the top left?**
BLIP3:None  Qwen2.5VL:No  InternVL2.5:No  GT:Porites

**Which coral genus occupies the most visible space?**
BLIP3:Porites  Qwen2.5VL:Porites  InternVL2.5:Porites  GT:Montipora

Figure 6: Selected examples of VQA results. Correct answers are shown in green and incorrect answers are shown in red. GT denotes ground truth.

## 2.4  Selected Examples of VQA Results

As shown in Figure 6, we present selected examples of VQA results from the visual-language models. In the first example, the coral reef image contains only one coral genus, with clearly defined taxonomic boundaries. The existing visual-language models exhibit reliable performance in answering the majority of questions. However, multiple coral genera in the images often display overlapping patterns or partial occlusion, resulting in unclear taxonomic boundaries among distinct genera. For instance, the third example demonstrates that current visual-language models show biases in understanding and responding to questions related to coral size and coral position.

## 2.5  Visualization of Coral Bleaching Regions

In our work, we have designed and performed experiments for assessing coral bleaching coverage, leading to the following findings. First, the mean absolute error (MAE) of Qwen2.5VL is 0.1124, whereas that of InternVL2.5 is 0.0818. The MASE for Qwen2.5VL and InternVL2.5 are 1.2326 and 0.8967, respectively. Second, compared with Qwen2.5VL, InternVL2.5 has lower values of MAE and MASE, indicating better generalize ability for complex reasoning tasks. Third, all methods still exhibit substantial prediction errors. Moreover, we are curious about the substantial prediction errors encountered in coral coverage evaluation. We visualize the bleaching regions of corals predicted by InternVL2.5. As shown in Figure 7, in the left image, the green areas represent the bleaching regions identified by InternVL2.5, while in the right image, the red and green areas indicate the bleaching regions manually annotated by experts. We can draw the following two conclusions. First, we find that the coral bleaching regions predicted by InternVL are substantially larger than the actual bleaching areas. Second, InternVL2.5 tends to misclassify reef substrates and dead corals as bleached corals, which further highlights the limitations of existing vision–language models in complex reasoning tasks.

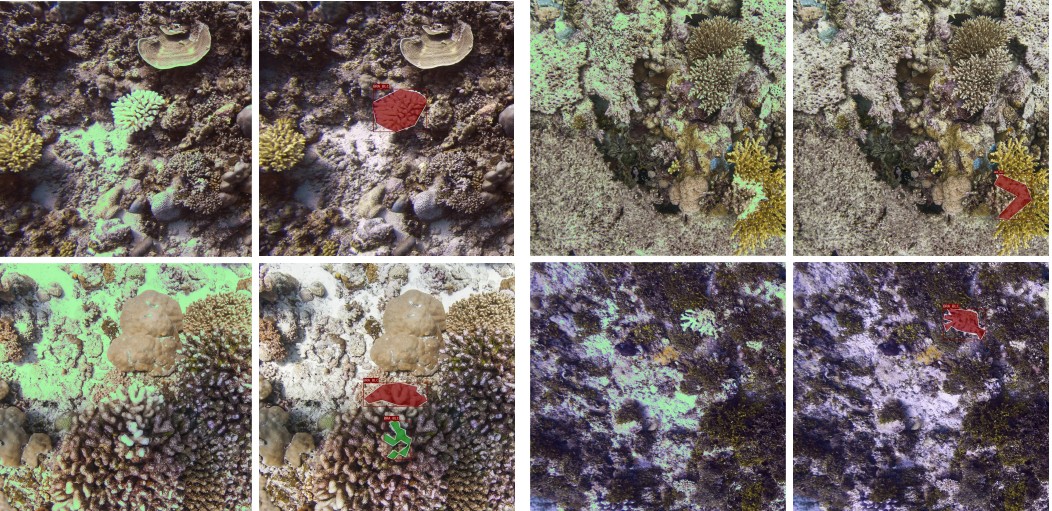

Figure 7: Visualization of coral bleaching regions predicted by InternVL2.5 and the corresponding expert-annotated ground truth.