# OpenReview forum: "CoralVQA: A Large-Scale Visual Question Answering Dataset for Coral Reef Image Understanding"
_NeurIPS.cc/2025/Datasets_and_Benchmarks_Track — NeurIPS 2025 Datasets and Benchmarks Track oral_

### Official Review · Reviewer_KfKN · 2025-06-13

**Rating:** 5
**Confidence:** 3

**Summary:**

This paper presents CoralVQA, a large-scale Visual Question Answering (VQA) dataset focused on coral reef imagery to support marine ecosystem monitoring and conservation efforts. The dataset includes 12,805 real-world coral images, annotated with 277,653 question–answer pairs spanning 16 visual and ecological dimensions. The authors propose a semi-automatic data construction pipeline, combining automated image processing and GPT-4o-based question generation, followed by expert verification. The paper also benchmarks several state-of-the-art large vision-language models (LVLMs) on CoralVQA, demonstrating the dataset’s complexity and the challenges it presents in domain-specific reasoning.

**Dataset Code Accessibility:**

Yes

**Ethical Considerations:**

No, there are no or only very minor ethics concerns

**Final Justification:**

Thanks for the author's rebuttal. Most of my concerns have been addressed. I keep my rating.

**Limitations Weaknesses:**

1. While powerful, the reliance on GPT-4o for generating Q&A pairs may introduce biases or inconsistencies not fully captured during human verification. The prompts and temperature settings could further influence the QA diversity.
2. While a case study is provided for coral size and quantity questions, deeper error analysis across other dimensions (e.g., symbiosis, bleaching susceptibility) would strengthen insights into model limitations.
3. The dataset would benefit from quantitative measures of annotation consistency or human performance benchmarks to contextualize model evaluations.

**Strengths Contributions:**

1. This is the first VQA dataset dedicated to coral reef understanding, addressing a critical gap in domain-specific multimodal research.
2. The dataset features 67 coral genera from 3 oceans, with questions designed across basic visual and ecological/health-related attributes, offering a rich and multi-faceted benchmark.
3. The authors propose a scalable semi-automatic pipeline, integrating marine biology expertise, GPT-4o for multimodal Q&A generation, and a rigorous human verification process.
4. The paper evaluates 4 prominent LVLMs (InternVL2.5, Qwen2.5VL, BLIP3, Mini-Gemini), providing detailed breakdowns by question type and generalization across regions and complex tasks (e.g., bleaching coverage estimation).
5. The dataset and code are publicly available on Hugging Face with clear documentation, supporting transparency and reuse.

---

> ### Author Rebuttal · Authors · 2025-07-31
>
> We would like to thank the reviewer's suggestions. We greatly appreciate the time and effort devoted to providing such detailed and valuable suggestions for enhancing our work.
>
> > Q1: While powerful, the reliance on GPT-4o for generating Q&A pairs may introduce biases or inconsistencies not fully captured during human verification. The prompts and temperature settings could further influence the QA diversity.
>
> We would like to thank the reviewer for raising the concerns of data quality. We agree that potential biases and inconsistencies introduced by GPT-4o in Q&A generation. To minimize the potential errors and biases, we took multiple measures as follows in the stage of question-answer generation. In this step, the major issue is about the biases and inconsistencies issues of GPT-4o generated question-answer pairs. To address the issue of inconsistencies, we set the lower temperature parameter of 0.3 and incorporate targeted prompt phrases that explicitly require definitive answers. We randomly sampled 1000 questions, with each question being answered 10 times by GPT-4o. The average number of consistent answers was 8.2 times. Since GPT4o's Q-A generation were largely consistent, and the thorough human verfication, the issue of potential biases and inconsistencies should be significant alleviated. As suggested by the reviewer, we will include a more detailed discussion of these limitations and our mitigation strategies in the revised manuscript.
>
> > Q2: While a case study is provided for coral size and quantity questions, deeper error analysis across other dimensions (e.g., symbiosis, bleaching susceptibility) would strengthen insights into model limitations.
>
> Thank you for this insightful suggestion. We agree that a more comprehensive error analysis would provide deeper insights into model limitations across different coral-related dimensions.
> We have expanded our error analysis including symbiosis relationships and bleaching susceptibility. This extended analysis reveals distinct failure patterns: models struggle particularly with symbiotic organism identification and bleached coral identification due to fine-grained visual similarities. we will include a more detailed analysis in the revised manuscript.
>
> > Q3: The dataset would benefit from quantitative measures of annotation consistency or human performance benchmarks to contextualize model evaluations.
>
> Thank you for this important suggestion regarding annotation quality assessment. As we mentioned in Section 3.6, we took a three-stage procedure to fix potential errors and provide the quantitative measures of annotation consistency. The first stage emplyed 12 students to verify the coral image and question-answer pairs manually and fix the errors resulted from using GPT4o. Each of them needed to deal with roughly 1,000 images (12,805/12 = 1,067.08). In total, there were 12 subsets of images and correspnding Q-A pairs. The second stage was cross-checking, each student cross-checked the subsect of images verfied by another student. If the consistency of cross-checking is lower than 95% in any specific subset, a third inspector was assigned to check these inconsistencies again and perform necessary fixes. In the final stage, domain experts inspected 10% images and corresponding Q-A pairs randomly choosen from each subset. If the accuracy of a subset's sample was lower than 95% (judged by the assigned expert), that subset had to be reverified again. After building the dataset, we asked three experts performed independent quality auditing. Each of them audited 500 iamges and corresponding Q-A pairs. All their ratings of accuracy were over 99% (on average, 99.47%), which means the error rate is about 0.5%. The consistency between annotators and experts was above 95% (with an average of 98.91%), which also indicates high annotation consistency. In addition, we will add a dedicated section presenting these quantitative measures and their implications for dataset quality.

---

### Official Review · Reviewer_qd2S · 2025-06-30

**Rating:** 5
**Confidence:** 4

**Summary:**

The authors propose CoralVQA, which they claim is the first large-scale Visual Question Aanswering (VQA) dataset dedicated to coral-reef images. The dataset comprises 12 805 underwater photographs from nine reef regions across three oceans annotated with 277 653 question–answer pairs spanning taxonomic, ecological, and domain-specific reasoning skills.
The authors detail a semi-automated pipeline that combines GPT-4o prompting with expert review to ensure accuracy. They release code, data, and baseline results for four leading Large Vision–Language Models (LVLMs), establishing benchmarks for future research in marine ecology and multimodal learning.

**Additional Feedback:**

Typos:
1. 5.3 “VQA Results on Cross-Region Dataset”, “Coral genera exhibit considerable intra-class varition …” should be “intra-class variation”
2. 5.1.2 “Coral VQA Tasks”, (3) “VQA on bleeching coverage dataset …” should be “bleaching coverage”

**Dataset Code Accessibility:**

Partly

**Dataset Code Comments:**

The Hugging Face repo already hosts images, JSON annotations and executable scripts, but the Croissant file is missing. The README is missing dependency installation instructions and inference examples.

**Ethical Considerations:**

No, there are no or only very minor ethics concerns

**Final Justification:**

Thanks for the authors' replies. In ny opinion, this manuscript can be accepted.

**Limitations Weaknesses:**

1 Insufficient overview of related work
The paper’s Introduction and Related-Work sections do not review existing marine or underwater VQA/multimodal datasets such as SeaVQA, FathomNet-QA, URPC-VQA, or Fish-VQA; instead, they cite only general-purpose VQA corpora (e.g., VQA v2, GQA) and a handful of coral classification/segmentation datasets.
As a result, readers cannot gauge how much CoralVQA advances beyond these “near-domain” efforts in scale and technical novelty. We recommend adding a systematic survey of the above underwater datasets, including a tabular comparison of image/QA size, taxonomic granularity, ocean-region coverage, and task types, and explaining how the coral-reef scenario amplifies or transforms known underwater challenges such as light attenuation, color cast, turbidity, and complex geometry. This would better highlight CoralVQA’s unique contributions and practical value.

2 Benchmark model diversity
Section 5.1 Implementation Details and Tables 3-5 report results for only four off-the-shelf, general-purpose LVLMs.
Because no model has been adapted or fine-tuned for marine or coral imagery, the study cannot quantify how much domain-specific knowledge would improve performance, leaving the true headroom above generic models unknown.
Please include at least one domain-aware baseline—for example, fine-tune an LVLM on an existing marine-life dataset (SeaVQA, FathomNet-QA, etc.) or adapt a coral-classification encoder (e.g., CoralNet) for VQA—to reveal the gains that specialized modelling might provide.

3 Cross-region split size
Table 2 shows the cross-region test set contains 994 images drawn solely from Hawai‘i.
This small, geographically limited sample makes it difficult to fully assess how models handle domain shifts across the world’s oceans.
We recommend expanding the Cross-Region split with additional samples from other areas—such as the Atlantic and Indian Oceans—to broaden test coverage and improve statistical reliability.

**Strengths Contributions:**

1. CoralVQA fills a clear gap: previous coral datasets focused only on classification or segmentation, and no public VQA resource existed.
2. The semi-automatic annotation workflow is transferable to other ecological domains.
3. Baselines for four LVLMs (InternVL 2.5, Qwen 2.5 VL, LLaVA-NeXT Yi, and InstructBLIP-13B) across three sub-tasks (in-domain, cross-region, and bleaching-coverage) provide a solid initial benchmark.
4. Paper is well-structured with clear figures of data pipeline and dataset taxonomy.
5. Potential for high impact in both ML benchmark research and coral-reef conservation applications.

---

> ### Author Rebuttal · Authors · 2025-07-31
>
> We would like to thank the reviewer's suggestions. We greatly appreciate the time and effort devoted to providing such detailed and valuable suggestions for enhancing our work.
>
> > Q1: Insufficient overview of related work The paper’s Introduction and Related-Work sections do not review existing marine or underwater VQA/multimodal datasets.
>
> Thank you for this excellent suggestion. We acknowledge that our related work section should be enhanced to cover existing marine and underwater VQA/multimodal datasets. We will expand the related work section to include a comprehensive review of marine-domain datasets including SeaVQA, FathomNet-QA, URPC-VQA, and Fish-VQA in the revised paper as suggested by the reviewer.
>
> > Q2:Benchmark model diversity Section 5.1 Implementation Details and Tables 3-5 report results for only four off-the-shelf, general-purpose LVLMs. Because no model has been adapted or fine-tuned for marine or coral imagery, the study cannot quantify how much domain-specific knowledge would improve performance, leaving the true headroom above generic models unknown. Please include at least one domain-aware baseline—for example, fine-tune an LVLM on an existing marine-life dataset (SeaVQA, FathomNet-QA, etc.) or adapt a coral-classification encoder (e.g., CoralNet) for VQA—to reveal the gains that specialized modelling might provide.
>
> Thank you for this valuable suggestion regarding benchmark diversity. We acknowledge that evaluating only general-purpose LVLMs limits our ability to quantify the potential gains from domain-specific adaptation. The FathomNet dataset cannot be directly applicable to VQA tasks. I have downloaded the FathomNet dataset and will build the FathomNet-QA dataset for domain-aware baseline—for example. In addition, in response to your recommendation, we have fine-tuned the Qwen2.5VL and InternVL2.5 on our CoralVQA dataset. As shown in the table, these fine-tuned models demonstrate substantial improvements over general-purpose LVLMs, with the fine-tuned InternVL2.5 achieving performance improvements of 43.68% and 35.54% on questions related to basic visual attributes and ecological and health-related attributes, respectively. These results reveal significant headroom above generic models and validate the importance of domain-specific knowledge for coral reef applications.
>
> Visual question answering performance of zero-shot and fine-tuned models on the test dataset：
> <div style="text-align: center;">
>
> | Methods        | Category | Presence | Quantity | Color | Position | Size | Shape | Scene | ALL  |
> |:--------------:| :----:   | :----:   | :----:   | :----:| :----:   |:----:| :----:| :----:|:----:|
> | Qwen2.5VL      | 63.72    | 51.55    | 12.59    | 53.05 | 47.45    | 58.59| 9.02  | 56.80 | 44.09|
> | InternVL2.5    | 23.94    | 54.39    | 13.22    | 35.19 | 8.03     | 4.62 | 18.88 | 63.11 | 27.67|
> | Qwen2.5VL(FT)  | 55.00    | 70.43    | 14.90    | 70.68 | 46.18    | 59.45| 57.47 | 94.70 | 58.60|
> | InternVL2.5(FT)| 81.69    | 79.45    | 35.54    | 78.27 | 62.18    | 74.46| 62.85 | 96.43 | 71.35|
> </div>
>
> <div style="text-align: center;">
>
> | Methods        |Growth    |Algal|Presence|Quantity|Integrity| Susceptibility| Environment|Symbiosis | ALL|
> |:--------------:| :----:   | :----:   | :----:   | :----:   | :----:  |:----:| :----:| :----:|:----:|
> | Qwen2.5VL      | 44.93    | 37.58    | 53.73    | 36.43    | 52.89   | 52.89| 68.63 | 87.62 | 54.34|
> | InternVL2.5    | 45.78    | 41.82    | 32.75    | 18.29    | 41.92   | 43.08| 61.49 | 81.93 | 45.88|
> | Qwen2.5VL(FT)  | 78.52    | 58.18    | 90.56    | 60.50    | 72.56   | 74.37| 72.00 | 72.45 | 72.39|
> | InternVL2.5(FT)| 86.96    | 65.45    | 88.73    | 65.25    | 95.90   | 80.04| 80.20 | 88.81 | 81.42|
> </div>
>
>
> > Q3:Cross-region split size Table 2 shows the cross-region test set contains 994 images drawn solely from Hawai‘i. This small, geographically limited sample makes it difficult to fully assess how models handle domain shifts across the world’s oceans. We recommend expanding the Cross-Region split with additional samples from other areas—such as the Atlantic and Indian Oceans—to broaden test coverage and improve statistical reliability.
>
> Thank you for this valuable feedback regarding the geographical diversity of our cross-region evaluation. Current cross-region test set of 994 images from Hawaii represents a limitation in assessing true global generalization capabilities. We incorporated samples from Atlantic and Indian Ocean regions to enhance the statistical robustness of our cross-region evaluation and provide a more comprehensive assessment of real-world domain adaptation capabilities across diverse marine ecosystems. We collected 2,288 coral reef images from the Indian Ocean and Chagos Archipelago, the Indian Ocean and Maldives to construct the Indian coral dataset. We sampled 926 coral reef images from Atlantic Ocean to build the Atlantic coral dataset. We gathered 1,607 coral reef images from the Pacific Ocean, Indonesia and Philippines to build the PAC_IDN_PHL dataset. More detailed cross-regional evaluation results will be provided in the revised paper.
>
> > Q4: Dataset Code Comments: The Hugging Face repo already hosts images, JSON annotations and executable scripts, but the Croissant file is missing. The README is missing dependency installation instructions and inference examples.
>
> We thank the reviewer for pointing out these issues. We will add the Croissant metadata file to ensure proper dataset discoverability and standardization. Additionally, we will provide a comprehensive README file containing detailed dependency installation instructions and practical inference examples.

---

### Official Review · Reviewer_dqXo · 2025-07-01

**Rating:** 4
**Confidence:** 4

**Summary:**

This  paper presents a large-scale VQA dataset of Coral Reef domain for MLLMs, which contains about 12,805 real-world images with about 277k QA pairs. The introduction of this dataset can well compensate the shortcoming of existing MLLMs in the domain of Coral Reef knowledge.

**Dataset Code Accessibility:**

Yes

**Dataset Code Comments:**

Not dataset code is given.

**Ethical Considerations:**

No, there are no or only very minor ethics concerns

**Final Justification:**

See review.

**Limitations Weaknesses:**

1. From Fig.1, most answers of these QA pairs are too easy to well facilitate the development of MLLMs. The design principle of this new dataset still follows the old one of classical VQA datasets like VQA2.0. However, for the development of MLLMs, more annotation content are welcome to better master the knowledge of this domain, such as CoT, dense captions or grounding information.

2. The quality of this dataset needs more supports. From Fig.2, the construction of this dataset still heavily rely on GPT-4o. Although it is a SOTA MLLMs with outstanding VL capability, its domain specific knowledge may be not good enough for correct annotations. Thus, how much efforts are devoted to the manual labeling should be clearly described.

**Strengths Contributions:**

1. The propose of this dataset can well provide the domain knowledge for MLLMs and address their shortcoming in the domain of Coral Reef.

2. The proposed dataset is relatively large enough, including 12k real-world collected images and more than 277k annotated QA pairs, which is a high contribution to the community.

3. The designs of question types are relatively reasonable, which can well covers the knowledge of this domain.

---

> ### Author Rebuttal · Authors · 2025-07-31
>
> We would like to thank the reviewer's suggestions. We greatly appreciate the time and effort devoted to providing such detailed and valuable suggestions for enhancing our work.
>
> > Q1: From Fig.1, most answers of these QA pairs are too easy to well facilitate the development of MLLMs. The design principle of this new dataset still follows the old one of classical VQA datasets like VQA2.0. However, for the development of MLLMs, more annotation content are welcome to better master the knowledge of this domain, such as CoT, dense captions or grounding information.
>
> Thank you for this insightful feedback. We acknowledge that our current QA pairs follow traditional VQA design principles, which limits their effectiveness for facilitate the development of MLLMs. We totally agree with the reviewer that more sophisticated annotations are important. However, our dataset is designed for filling the gap of lacking high-quality VQA dataset of coral reef, which is the first yet crucial step towards more sophisticated datasets and models such as MLLMs. We will include a discussion of this limitation in the revised manuscript and propose incorporating CoT reasoning, dense captions, and grounding information as key future directions for dataset enhancement. These additions would indeed better facilitate MLLM development in specialized domains.
> Your valuable suggestion will guide our future work toward creating more comprehensive benchmarks for domain-specific MLLMs.
>
> > Q2: The quality of this dataset needs more supports. From Fig.2, the construction of this dataset still heavily rely on GPT-4o. Although it is a SOTA MLLMs with outstanding VL capability, its domain specific knowledge may be not good enough for correct annotations. Thus, how much efforts are devoted to the manual labeling should be clearly described.
>
> We would like to thank the reviewer for raising the concerns of data quality. We acknowledge that using GPT-4o for question-answer generation and answer matching may indeed introduce a certain degree of bias and issues. While Figure 2 shows GPT-4o's involvement in Q&A generation, we want to emphasize that substantial manual effort was invested throughout the pipeline. To obtain accurate question-answer pairs, we devoted more effort to label cleaning and re-annotating,  and human verification. To minimize the potential errors and biases, we took multiple measures as follows.
>
> Step 2--Label cleaning and re-annotating
>
> In this step, the major issue is inconsistent annotation standards and incorrect coral annotations. To address the problem, we have every image re-examined by three marine scientists after the label cleaning and re-annotating step, and adopt a majority voting approach to determine the final annotations. In fact, only a very small set of images (203 out of 12,805, <2%) contains errors.
>
> Step 6--Human verification
>
> This step was designed to fix errors with GPT-4o generated question-answer pairs. As we mentioned in Section 3.6, we took a three-stage procedure to fix potential errors. The first stage emplyed 12 students to verify the coral image and question-answer pairs manually and fix the errors resulted from using GPT4o. Each of them needed to deal with roughly 1,000 images (12,805/12 = 1,067.08). In total, there were 12 subsets of images and correspnding Q-A pairs. The second stage was cross-checking, each student cross-checked the subsect of images verfied by another student. If the consistency of cross-checking is lower than 95% in any specific subset, a third inspector was assigned to check these inconsistencies again and perform necessary fixes. In the final stage, domain experts inspected 10% images and corresponding Q-A pairs randomly choosen from each subset. If the accuracy of a subset's sample was lower than 95% (judged by the assigned expert), that subset had to be reverified again.
>
> While we took such steps to deal with potential errors and biases, we cannot gurantee the dataset is bug-free. After building the dataset, we asked three experts performed independent quality auditing. Each of them audited 500 iamges and corresponding Q-A pairs. All their ratings of accuracy were over 99% (on average, 99.47%), which means the error rate is about 0.5%.
>
> Through these procedures, the subjectivity should be well controlled since we integrated views from multiple annotators and experts. Meanwhile, since GPT4o's Q-A generation were largely consistent, and the thorough human verfication, the issue of reproductivity should be significant alleviated. As suggested by the reviewer, we will add more detailed description of the quality control measures and enhance the discussion of the related subjectivity and reproductivity issues.

---

### Official Review · Reviewer_MFCz · 2025-07-06

**Rating:** 6
**Confidence:** 4

**Summary:**

This manuscript introduces a new VQA dataset related to the Coral reef animal. The dataset is exquisitely constructed and well tested, showcasing its importance and usefulness for future ML / marine biology research.

**Additional Feedback:**

see weaknesses

**Dataset Code Accessibility:**

Yes

**Dataset Code Comments:**

Looks good on huggingface to me

**Ethical Considerations:**

No, there are no or only very minor ethics concerns

**Final Justification:**

I'm nearly shocked, the closed-source evals have been ran and reported! It would be a shame if this paper does not get at least a spotlight.

**Limitations Weaknesses:**

Minor only:
- Line 29 "Core reef"
- Please mention that the training details are in the supplement, I wanted to know the details and had to guess they were there.
- section 5.4 would be easier to understand with a figure showing the failure of the region masks (I think the table could be merged into the text). Also, you should define the MAE/MASE are (just in case someone doesn't already know)
- If you have the time/money, I would be interested to see how closed-source models perform on the test set.
- I'm not sure what Figure 5 is referencing. Section 5.5 I assumed

**Strengths Contributions:**

The paper was a joy to read. The writing is extremely clear and easy to follow. Here's a random (incomplete) assortment of strengths:

- Important problem: Coral reef health being a key indicator of global ocean ecological health
- Clear motivation: Current datasets do not match the LVLM structure of modern AI
- Incredible data pipeline: What a massive amount of work for the original labels!
- Sensible VQA generation: I find the gpt4o use very in line with other methods for creating such a large corpus of VQA data
- Human-in-the-loop sanity checks! You would not believe how often datasets skip this step
- Evals by finetuning models: I love it, many datasets are just eval only without training. All the evals are exactly constructed how I would expect, very sensible the whole way through.
- The additional experiments from 5.4 and 5.5 are great extensions.

Overall a stellar paper. You can easily see the care and passion that went into it.

---

> ### Author Rebuttal · Authors · 2025-07-31
>
> We would like to thank the reviewer's suggestions. We greatly appreciate the time and effort devoted to providing such detailed and valuable suggestions for enhancing our work.
>
> > Issue 1: Language problems and typos
>
> We will fix the lanaguage problems and typos when revising the paper.
>
> > Issue 2:  Please mention that the training details are in the supplement, I wanted to know the details and had to guess they were there.
>
> Yes, they are in the supplement. We will clarify it in the main text and point it to the related file in the supplement.
>
> > Issue 3: Section 5.4 would be easier to understand with a figure showing the failure of the region masks (I think the table could be merged into the text). Also, you should define the MAE/MASE are (just in case someone doesn't already know).
>
> This is a great suggestion. We will add a figure illustrating the failure cases of region masks and integrate the table into the text in the revised paper. Furthermore, we will provide the detailed definitions for MAE and MASE metrics.
>
> > Issue 4: How closed-source models perform on the test set.
>
> We perform some new experiments with two closed-source models: GPT-4o and Claude-3.5 Haiku. As shown in the table, although existing closed-source models have achieved good performance on general tasks, they perform poorly on coral-related tasks, which demonstrates the necessity of coral VQA data and model fine-tuning. We will include results from additional baseline LVLMs in the revised paper.
>
> Visual question answering performance of closed-source models on the test dataset
> <div style="text-align: center;">
>
> | Methods        | Category | Presence | Quantity | Color | Position | Size | Shape | Scene | ALL  |
> |:--------------:| :----:   | :----:   | :----:   | :----:| :----:   |:----:| :----:| :----:|:----:|
> | GPT-4o         | 71.35    | 50.38    | 7.04     | 70.09 | 43.13    | 54.26| 57.47 | 85.83 | 54.94|
> | Claude3.5 Haiku| 69.27    | 52.26    | 5.92     | 43.08 | 48.37    | 62.63| 38.36 | 47.54 | 45.93|
> | Qwen2.5VL(FT)  | 55.00    | 70.43    | 14.90    | 70.68 | 46.18    | 59.45| 57.47 | 94.70 | 58.60|
> | InternVL2.5(FT)| 81.69    | 79.45    | 35.54    | 78.27 | 62.18    | 74.46| 62.85 | 96.43 | 71.35|
> </div>
> <div style="text-align: center;">
>
> | Methods        |Growth    |Algal     |Presence  |Quantity  |Integrity| Susceptibility| Environment|Symbiosis | ALL|
> |:--------------:| :----:   | :----:   | :----:   | :----:   | :----:  |:----:| :----:| :----:|:----:|
> | GPT-4o         | 52.81    | 42.88    | 25.21    | 14.78    | 33.68   | 28.68| 61.84 | 87.48 | 43.42|
> | Claude3.5 Haiku| 44.00    | 38.48    | 42.96    | 34.11    | 18.05   | 30.64| 62.00 | 30.64 | 37.61|
> | Qwen2.5VL(FT)  | 78.52    | 58.18    | 90.56    | 60.50    | 72.56   | 74.37| 72.00 | 72.45 | 72.39|
> | InternVL2.5(FT)| 86.96    | 65.45    | 88.73    | 65.25    | 95.90   | 80.04| 80.20 | 88.81 | 81.42|
> </div>
>
> > Issue 5: Figure 5 is Referencing figure 5.5.
>
> Thank you for pointing out this confusion. Yes, Figure 5 should be referenced in Section 5.5. We will fix it in the revised manuscript.

---

### Official Review · Reviewer_bLiz · 2025-07-09

**Rating:** 5
**Confidence:** 4

**Summary:**

This work introduces CoralVQA, the first large-scale coral reef visual question answering (VQA) dataset containing more than 277K images involving 12K images covering basic visual attributes and ecological and health-related conditions. To prepare CoralVQA, a semi-automatic data construction pipeline was developed in collaboration with marine biologists. Analysis of 4 LVLMs over different evaluation settings of CoralVQA exposes limitations and opportunities for building domain-specific LVLMs for coral reef understanding.

**Dataset Code Accessibility:**

Yes

**Ethical Considerations:**

No, there are no or only very minor ethics concerns

**Final Justification:**

The authors have addressed my concerns by adding a series of new experiments and clarification details about the accuracy and protocols followed for each step. The authors are strongly encouraged to include the additional details of the quality control measures and the new experimental results in the revised version of this paper.

**Limitations Weaknesses:**

1. A major limitation of this work is the limited discussion of the subjectivity and lack of reproducibility of the various steps of the semi-automatic data construction pipeline. Are there any annotation errors in the six steps (e.g.,  in Human Verification)? How is annotator bias handled? Further details reporting the number of times re-evaluation was performed based on sample quality will help to understand the errors in human verification. It will also be good to understand if there are any errors that remain in CoralVQA after the manual processing steps or is every image and QA pair manually verified for correctness.

2. The use of GPT4o for question answer generation and answer matching can introduce issues. Does it not bias the performance of GPT4o on this dataset, which although has not been included in this work, can be explored in the future? Is it possible to quantify how many times the responses of GPT4o was manually pruned, to understand the limitations of GPT4o? If manual verification was performed on the generated responses of the open-ended questions instead of using GPT4o, how do the results vary?

3. It is mentioned that "On average, each coral reef image has 21.6 questions." But there are only 16 question categories. Why not consider all 16 categories exhaustively for every image, resulting in 204,880 QA pairs? Is there a value in repeating the same question more than once for a given image?

4. The choice of baseline LVLMs needs justification. Why only 4 models considered, when there are many other state-of-the-art choices including GPT4o, Llama, Claude, Molmo, etc. Some of these models can be tested for zero-shot effectiveness too. Including a broader set of LVLMs will help establish the significance of the results beyond simplistic baselines.

5. The discussion section shows that LVLMs are failing at identifying the coral coordinates for an example image. Is it possible to test teh localization abilities of LVLMs explicitly as a separate task (e.g., by checking for the accuracy in generating coral coordinates)? Other tests can be performed to evaluate reasoning abilities of LVLMs including tests of hallucination.

6. For textual attribute extraction. it is mentioned that automated scripts were used based on the existing annotation files. What are these files?

7. It is mentioned that "Our pipeline can be widely applied to other marine domains, such as visual reasoning tasks in mangrove ecosystem imagery." Is there any evidence to support this claim?

**Strengths Contributions:**

1. Solves an impactful real-world problem and the approach is well-motivated.
2. Contributions are clearly stated.
3. Every step of the semi-automatic pipeline is described in detail.

---

> ### Author Rebuttal · Authors · 2025-07-31
>
> We would like to thank the reviewer's suggestions. We greatly appreciate the time and effort devoted to providing such detailed and valuable suggestions for enhancing our work.
>
> > Q1:The limited discussion of the subjectivity and lack of reproducibility of the various steps of the semi-automatic data construction pipeline.
>
> We would like to thank the reviewer for raising the concerns of data quality and reproducibility. We agree that errors and biases can occur in the semi-automatic pipeline. Among the six steps, three steps are particulerly error-prone, which are: label cleaning and re-annotating, question-answer generation, and human verification. To minimize the potential errors and biases, we took multiple measures as follows.
>
> Step 2--Label cleaning and re-annotating
>
> In this step, the major issue is inconsistent annotation standards and incorrect coral annotations. To address the problem, we have every image re-examined by three marine scientists after the label cleaning and re-annotating step, and adopt a majority voting approach to determine the final annotations. In fact, only a very small set of images (203 out of 12,805, <2%) contains errors.
>
> Step 5--Question-answer generation
>
> In this step, the major issue is about the reproducibility issues of GPT-4o generated question-answer pairs. To address the issue, we set the lower temperature parameter of 0.3 and incorporate targeted prompt phrases that explicitly require definitive answers. We randomly sampled 1000 questions, with each question being answered 10 times by GPT-4o. The average number of consistent answers was 8.2 times. Although GPT-4o generated answers exhibit minor inconsistencies, since all question answers undergo multiple rounds of manual verification, the impact of answer inconsistencies is further mitigated.
>
> Step 6--Human verification
>
> This step was designed to fix errors with GPT-4o generated question-answer pairs. As we mentioned in Section 3.6, we took a three-stage procedure to fix potential errors. The first stage employed 12 students to verify the coral image and question-answer pairs manually and fix the errors resulted from using GPT4o. Each of them needed to deal with roughly 1,000 images (12,805/12 = 1,067.08). In total, there were 12 subsets of images and correspnding Q-A pairs. The second stage was cross-checking, each student cross-checked the subsect of images verfied by another student. If the consistency of cross-checking is lower than 95% in any specific subset, a third inspector was assigned to check these inconsistencies again and perform necessary fixes. In the final stage, domain experts inspected 10% images and corresponding Q-A pairs randomly choosen from each subset. If the accuracy of a subset's sample was lower than 95% (judged by the assigned expert), that subset had to be reverified again.
>
> While we took such steps to deal with potential errors and biases, we cannot gurantee the dataset is bug-free. After building the dataset, we asked three experts performed independent quality auditing. Each of them audited 500 iamges and corresponding Q-A pairs. All their ratings of accuracy were over 99% (on average, 99.47%), which means the error rate is about 0.5%.
>
> Through these procedures, the subjectivity should be well controlled since we integrated views from multiple annotators and experts. Meanwhile, since GPT4o's Q-A generation were largely consistent, and the thorough human verfication, the issue of reproductivity should be significant alleviated. As suggested by the reviewer, we will add more detailed description of the quality control measures and enhance the discussion of the related subjectivity and reproductivity issues.
>
> > Q2: Using GPT-4o for both question generation and answer matching may bias its performance on this dataset. Could you quantify how often GPT-4o responses required manual pruning?
>
> Thank you for raising the concern about potential bias introduced by using GPT-4o for both question generation and answer matching. Based on the reviewers' suggestions, we compiled statistics on the manual pruning of GPT-4o responses. During the human verification stage, we modified 13.4% of the questions generated by GPT-4o and 56.8% of the answers generated by GPT-4o required manual pruning.
>
> > Q3:You mention an average of 21.6 questions per image across 16 categories. Why not use all 16 categories per image (yielding 204,880 QA pairs) rather than repeating questions? What's the value of question repetition per image?
>
> Thank you for raising questions regarding the number of question-answer pairs. Yes, there are 16 categories of questions for each image. However, a category may contains more than one questions. For example, a coral in a image often belongs to multiple coral genera, thereby the image may associated with multiple non-repetitive questions in the coral genera category. Therefore, each image may have more than 16 questions, making the total number of Q-A pairs is 277,653 (12,805  $\times$ 21.3), which is well beyond 204,880 (12,805 $\times$ 16).
>
> > Q4: Including more LVLMs would better establish result significance beyond basic baselines.
>
> Thank you for your valuable feedback. We acknowledge the importance of expanding the scope of baseline model comparisons. We perform some new experiments with two closed-source models: GPT-4o and Claude-3.5 Haiku. As shown in the table, although existing closed-source models have achieved good performance on general tasks, they perform poorly on coral-related tasks, which demonstrates the necessity of coral VQA data and model fine-tuning. We will include results from additional baseline LVLMs in the revised paper.
>
> Visual question answering performance of closed-source models  on the test dataset
> <div style="text-align: center;">
>
> | Methods        | Category | Presence | Quantity | Color | Position | Size | Shape | Scene | ALL  |
> |:--------------:| :----:   | :----:   | :----:   | :----:| :----:   |:----:| :----:| :----:|:----:|
> | GPT-4o         | 71.35    | 50.38    | 7.04     | 70.09 | 43.13    | 54.26| 57.47 | 85.83 | 54.94|
> | Claude3.5 Haiku| 69.27    | 52.26    | 5.92     | 43.08 | 48.37    | 62.63| 38.36 | 47.54 | 45.93|
> | Qwen2.5VL(FT)  | 55.00    | 70.43    | 14.90    | 70.68 | 46.18    | 59.45| 57.47 | 94.70 | 58.60|
> | InternVL2.5(FT)| 81.69    | 79.45    | 35.54    | 78.27 | 62.18    | 74.46| 62.85 | 96.43 | 71.35|
> </div>
> <div style="text-align: center;">
>
> | Methods        |Growth    |Algal     |Presence  |Quantity  |Integrity| Susceptibility| Environment|Symbiosis | ALL|
> |:--------------:| :----:   | :----:   | :----:   | :----:   | :----:  |:----:| :----:| :----:|:----:|
> | GPT-4o         | 52.81    | 42.88    | 25.21    | 14.78    | 33.68   | 28.68| 61.84 | 87.48 | 43.42|
> | Claude3.5 Haiku| 44.00    | 38.48    | 42.96    | 34.11    | 18.05   | 30.64| 62.00 | 30.64 | 37.61|
> | Qwen2.5VL(FT)  | 78.52    | 58.18    | 90.56    | 60.50    | 72.56   | 74.37| 72.00 | 72.45 | 72.39|
> | InternVL2.5(FT)| 86.96    | 65.45    | 88.73    | 65.25    | 95.90   | 80.04| 80.20 | 88.81 | 81.42|
> </div>
>
> > Q5:The discussion section shows that LVLMs are failing at identifying the coral coordinates for an example image. Is it possible to test teh localization abilities of LVLMs explicitly as a separate task (e.g., by checking for the accuracy in generating coral coordinates)? Other tests can be performed to evaluate reasoning abilities of LVLMs including tests of hallucination.
>
> Thank you for your valuable suggestion. Evaluating the localization capabilities of LVLMs as an independent task is a valuable research direction. Our CoralVQA dataset represents a first and fundamental step that provides the foundation for constructing localization datasets. We plan to build the localization dataset and include localization accuracy evaluation experiments in the revised manuscript to further evaluate reasoning abilities of LVLMs including tests of hallucination.
> > Q6:What are the existing annotation files mentioned for automated textual attribute extraction?
>
> Thank you for your valuable suggestion. These files contain detailed attribute information for each coral in the images, including categories, coordinates, health status, and other properties. I have released these files through the data link provided in the paper. We added a new "Read Me" file in the data package to provide explain these files.
>
> > Q7:Is there evidence for the claimed broader marine domain applicability?
> It is mentioned that "Our pipeline can be widely applied to other marine domains, such as visual reasoning tasks in mangrove ecosystem imagery." Is there any evidence to support this claim?
>
> Thank you for your valuable suggestion. Both coral reef images and mangrove images possess complex branching structures and rich surface texture features. Sauder[1] and Giardino[2] indicate that the proposed methods can be extended to underwater ecosystems of mangroves. Therefore, our pipeline can be used to create datasets related to mangrove ecosystem imagery.
>
> [1] Sauder, J., Banc‐Prandi, G., Meibom, A., & Tuia, D. (2024). Scalable semantic 3D mapping of coral reefs with deep learning. Methods in Ecology and Evolution, 15(5), 916-934.
>
> [2] Giardino, C., Bresciani, M., Fava, F., Matta, E., Brando, V. E., & Colombo, R. (2015). Mapping submerged habitats and mangroves of Lampi Island Marine National Park (Myanmar) from in situ and satellite observations. Remote Sensing, 8(1), 2.

---

> > ### Comment · Reviewer_bLiz · 2025-08-09
> > **Thank you**
> >
> > Thank you for your very thoughtful response to my comments. All my concerns  seem to have been addressed. The authors are strongly encouraged to include the new details and experiments in the revised version of the paper.

---

> > > ### Author Response · Authors · 2025-08-09
> > >
> > > We sincerely appreciate your thoughtful review and encouraging remarks. We are glad that our revisions have addressed your concerns, and we will make sure to include all the new details and experiments in the revised manuscript as suggested.

---

### Comment · Area_Chair_bj5b · 2025-08-01
**Reviewer response to rebuttals**

Reviewers, can you please take a look at the author's rebuttal and respond as soon as possible.

---

> ### Comment · Reviewer_MFCz · 2025-08-04
> **Maybe Oral quality**
>
> I'm in love with this paper. It's the best dataset paper I've read in a long time. I absolutely believe this paper should get a spotlight. I also could easily see it being an Oral (I'm not sure how it'd rank versus other spotlights)

---

> > ### Author Response · Authors · 2025-08-05
> >
> > We are glad you enjoy our work! It is a great honor for us. We're committed to making both the paper and dataset better based on the feedback and continue our AI-enabled coral conservation efforts.

---

### Decision · Program_Chairs · 2025-09-18

**Decision:**

Accept (oral)

**Comment:**

All the reviewers and the AC agree that this benchmark is a strong contribution and recommend acceptance, citing that the benchmark "solves an impactful real-world problem," the documentation of the benchmark curation process was comprehensive and clear, and the evaluation and analysis highlighted interesting dimensions of future work for the research community.